# Lysophosphatidic acid modulates CD8 T cell immunosurveillance and metabolism to impair anti-tumor immunity

Jacqueline A. Turner [1,2], Malia A. Fredrickson [1], Marc D'Antonio[1], Elizabeth Katsnelson[3], Morgan MacBeth [4], Robert Van Gulick[4], Tugs-Saikhan Chimed[4], Martin McCarter[3], Angelo D'Alessandro[5], William A. Robinson[4], Kasey L. Couts[4], Roberta Pelanda[1], Jared Klarquist[1], Richard P. Tobin[3] & Raul M. Torres [1] ✉

Lysophosphatidic acid (LPA) is a bioactive lipid which increases in concentration locally and systemically across different cancer types. Yet, the exact mechanism(s) of how LPA affects CD8 T cell immunosurveillance during tumor progression remain unknown. We show LPA receptor (LPAR) signaling by CD8 T cells promotes tolerogenic states via metabolic reprogramming and potentiating exhaustive-like differentiation to modulate anti-tumor immunity. We found LPA levels predict response to immunotherapy and Lpar5 signaling promotes cellular states associated with exhausted phenotypes on CD8 T cells. Importantly, we show that Lpar5 regulates CD8 T cell respiration, proton leak, and reactive oxygen species. Together, our findings reveal that LPA serves as a lipid-regulated immune checkpoint by modulating metabolic efficiency through LPAR5 signaling on CD8 T cells. Our study offers key insights into the mechanisms governing adaptive anti-tumor immunity and demonstrates LPA could be exploited as a T cell directed therapy to improve dysfunctional anti-tumor immunity.

Lysophosphatidic acid (LPA) is a bioactive, pleiotropic lipid mediator with functions important for the development and progression of cancer[1]. Although LPA can be generated intracellularly, the vast majority of bioactive LPA is synthesized extracellularly by the secreted ectoenzyme, autotaxin (ATX, gene name *ENPP2*), a phospholipase D that is secreted and associates with integrins on the plasma membrane surface[2–4]. LPA molecular species exist as saturated or unsaturated 14, 16, 18, 20, or 22 carbon chains. Extracellular circulating LPA may bind to one of six LPA receptors (LPARs) which are G protein coupled receptors (GPCRs), denoted LPAR1-6[5,6]. These LPARs may signal through $G_{\alpha 12/13}$, $G_{\alpha q/11}$, $G_{\alpha i/o}$, and $G_{\alpha s}$ to downstream effectors

including Rho, PLC/IP₃/DAG, MAPK, PI3K/ATK, and adenylyl cyclase/cAMP[2]. LPA has numerous effects which are cell-type specific and LPAR dependent.

There is increasing evidence that metabolism and lipid signaling play important roles in regulating T cell fate, differentiation, and effector function[6–8]. Dysfunctional CD8 T cells are associated with specific metabolic states[9,10]. Particularly, oxidative cell stress and impaired mitochondrial oxidative phosphorylation are repeatedly found to be associated with T cell dysfunction[11,12]. Fatty acid oxidation has been shown to be important in CD8 T cell differentiation, recall capacity, and interferon γ (IFNγ) production[11,13–15]. These CD8 T cell

[1]Department of Immunology and Microbiology, University of Colorado School of Medicine, Anschutz Medical Campus, Aurora, CO, USA. [2]Medical Scientist Training Program, University of Colorado School of Medicine, Anschutz Medical Campus, Aurora, CO, USA. [3]Division of Surgical Oncology, Department of Surgery, University of Colorado School of Medicine, Anschutz Medical Campus, Aurora, CO, USA. [4]Division of Medical Oncology, Department of Medicine, University of Colorado School of Medicine, Anschutz Medical Campus, Aurora, CO, USA. [5]Department of Biochemistry and Molecular Genetics, University of Colorado School of Medicine, Anschutz Medical Campus, Aurora, CO, USA. ✉e-mail: raul.torres@cuanschutz.edu

functions are critical for mounting anti-tumor immune responses. Thus, modulating metabolism specific to T cells could serve as an effective mechanism to rescue T cell function.

In this work, we investigate how LPA modulates CD8 T cell metabolism, function, and phenotype. Previous work from our lab has demonstrated that naïve and effector T cells express LPARs 2/5/6 and LPAR5 negatively regulates CD8 T cell receptor signaling and effector function by co-opting the cell cytoskeleton during immune synapse formation[8,16]. Since metabolic dysfunction in CD8 T cells, impaired antigen-specific killing, and poor responses to immunotherapy are characteristics of CD8 T cells exhaustion, we hypothesized and tested whether LPA and Lpar5 signaling modulates CD8 T cell metabolic fitness and dysfunctional phenotypes. Our findings describe a role for Lpar5 in CD8 T cells to promote exhaustive-like differentiation and modulate metabolic fitness. We propose LPAR5 serves as a lipid-regulated immune checkpoint which impairs anti-tumor immunity through multiple mechanisms that include metabolic reprogramming of tumor-specific CD8 T cells and direct inhibition of antigen receptor-induced T cell activation. These findings provide strong evidence that Lpar5 signaling is a targetable T cell directed therapy for improving endogenous immune responses in cancer.

## Results

### LPA and ATX correlate with an exhausted-like T cell phenotype and patient outcomes

LPA has been reported to have many pro-tumorigenic properties[5,6,17,18], thus we initially examined the role of LPA signaling in humans by hypothesizing that ATX and LPA may serve as prognostic indicators of disease outcome in cancer. To investigate whether ATX correlates with tumor progression and patient outcome, we examined all curated non-redundant studies from cBioportal[19,20] for progression free survival. It is well established that the proto-oncogene *MYC* is amplified in many tumors and is associated with poor outcomes[21]. Thus, we used *MYC* amplification as a comparison group and stratified patients with tumors amplified for *ENPP2*, *MYC*, or wildtype for both *ENPP2* and *MYC*. These analyses revealed that patients with *ENPP2* amplification had a significantly worse progression free survival even when compared to patients with *MYC* amplification (Fig. 1A). Notably, there were diverse types of cancers that amplify *ENPP2*, and importantly there was not a dominating overrepresentation of one cancer type in the patient cohort (Supplementary Fig. 1A–F).

To assess *ENPP2* expression, we correlated melanoma TCGA *ENPP2* mRNA with a previously generated cytotoxic T lymphocyte signature[22] and exhaustion markers[23] to generate an "exhaustion" signature (Fig. 1B, Source Data, Supplementary Fig. 1G). We found melanoma tumors with high *ENPP2* expression were enriched for an "exhausted" CD8 T cell profile by transcriptomic gene analysis. Previously, we have shown that LPAR5-deficient CD8 T cells are better able to kill melanoma tumor cells in vitro and control local tumor growth after implantation and compared to wildtype CD8 T cells[8,16]. The first report of ATX generating LPA was first identified in melanoma and our laboratory has established interest in examining the role of ATX in melanoma[24], as such, we specifically focused on this cancer type for our study. Using previously published single cell RNA sequencing data[25], we determined *LPAR5* is not expressed by most melanoma tumor cells but is expressed by immune cells including CD8 T cells (Fig. 1C, D). Further, *LPAR5* expression decreases with increasing tumor purity (Supplementary Fig. 1H) consistent with *LPAR5* expression predominantly restricted to immune cells. Based on these findings, we performed a correlation analysis of *LPAR5* expression versus our "exhaustion" signature and found a strong correlation between *LPAR5* expression and markers associated with CD8 T cell exhaustion (Fig. 1E). These data implicate *ENPP2* and *LPAR5* as negative prognostic factors modulating CD8 T cell phenotypes and anti-tumor immunity.

We next aimed to examine the role of LPA in predicting outcomes in our own patient cohort. We performed lipidomics on plasma samples from stage IV melanoma patients to measure LPA pre- and post-immunotherapy treatment. We determined that 16:0 LPA is detected at significantly lower concentrations in stage IV melanoma patients who respond to immunotherapy compared to non-responder patients (Fig. 1F, Source Data). Notably, 16:0 and 18:1 are the most abundant plasma LPA species observed[26]. We also measured 16:1, 18:0, 18:1, 18:2, 20:4, and 22:6 LPA species pre- and post-immunotherapy (Supplementary Fig. 2) and while we observed similar trends that non-responders harbored elevated LPA, none of the other LPA species were found to be significantly different between the responders and non-responders. These data indicate 16:0 molecular species of LPA is a specific biomarker of T cell phenotype and predicts response to immunotherapy.

### Lpar5 signaling on CD8 T cells impairs anti-tumor immunity in vivo

Our laboratory has previously shown LPA signaling through LPAR5 impairs T cell activation and cytotoxic activity[8,16]. To understand the role of LPA in the context of anti-tumor immunity, we performed adoptive transfers using a systemic tumor model with the hypothesis that Lpar5 receptor knockout (*Lpar5⁻/⁻*) CD8 T cells also display an enhanced ability to kill tumor cells systemically in vivo. OT-I is a transgenic T cell receptor with specificity for a chicken ovalbumin peptide (SIINFEKL or N4)[27]. Thus, in these experiments OT-I CD8 T cells serve as surrogate tumor-specific T cells. We intravenously injected syngeneic mouse melanoma B16 cells expressing chicken ovalbumin (B16.cOVA, surrogate tumor antigen) into wildtype C57BL/6 hosts and on the same day adoptively transferred wildtype or *Lpar5⁻/⁻* OT-I CD8 T cells into the same host mice. (Fig. 2A). After twenty days, recipient mice adoptively transferred with *Lpar5⁻/⁻* OT-I CD8 T cells had fewer, smaller, and more circumscribed tumors when compared to hosts receiving wildtype OT-I CD8 T cells (Fig. 2B–E). Mice receiving receptor-deficient OT-I CD8 T cells also had more tumor infiltrating CD8⁺ cells and expressed fewer markers associated with exhaustion compared to wildtype recipients (Fig. 2F–H, Supplementary Fig. 3). Overall, *Lpar5⁻/⁻* OT-I CD8 T cells responding to melanoma in vivo display fewer inhibitory receptors and improved anti-tumor immunity.

### Lpar5 signaling drives exhaustive-like differentiation on CD8 T cells in vitro and in vivo

Since we observed decreased Tim3 expression on PD1⁺ *Lpar5⁻/⁻* OT-I CD8 T cells isolated from melanoma tumors compared to wildtype OT-I CD8 T cells (Fig. 2H), we sought to further investigate how LPA signaling might modulate exhausted and/or dysfunctional phenotypes. To accomplish this, we treated OT-I effector CD8 T cells with LPA in the presence or absence of chronic TCR stimulation in vitro (Fig. 3A). Of note, longer-term in vitro cultures necessitate the use of (fetal bovine) serum which contains low levels of LPA[28] that likely signal via Lpar5 throughout this culture period; nevertheless, we supplemented with additional LPA to our cultures to ensure sustained LPA exposure for this prolonged in vitro assay. Both OT-I and *Lpar5⁻/⁻* OT-I CD8 T cell cultures treated with anti-CD3 + LPA resulted in virtually all CD8 T cells to dually express PD1 and Tim3 (Fig. 3B–D, Supplementary Fig. 4) although the level of these inhibitory receptors were reduced on *Lpar5⁻/⁻* OT-I CD8 T cells (Fig. 3E, H, Supplementary Fig. 4). Interestingly, LPA supplementation alone in cultures of *Lpar5⁻/⁻* OT-I CD8 T cells also resulted in a significantly decreased percent of PD1⁺ Tim3⁺ compared to wildtype OT-I cells (Fig. 3B–D, Supplementary Fig. 4). In line with our previous findings, we observed that *Lpar5⁻/⁻* OT-I effector CD8 T cells that were chronically stimulated expressed less PD1 and Tim3 (Fig. 3E–J). Given the robust differences we observed in vivo, we chose to further investigate how LPA and Lpar5 signaling modulates exhaustion phenotypes using Lpar5 knockout mice and in vivo models.

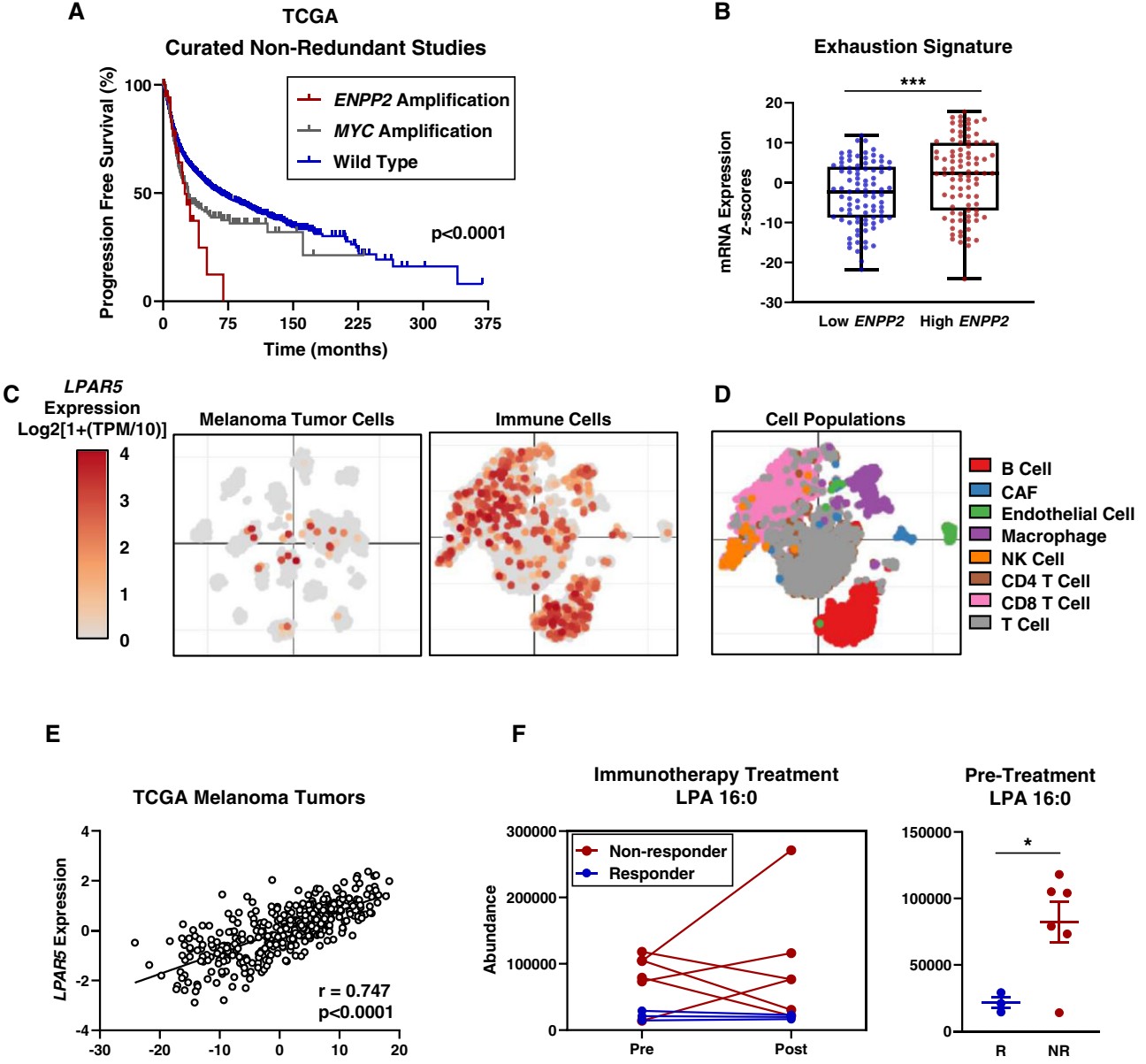

**Fig. 1 | Lysophosphatidic acid is a prognostic marker in solid tumor malignancies. A** Analysis of data from The Cancer Genome Atlas (TCGA) on progression free survival. Data was taken as pan-cancer data from all solid tumors in cBioPortal from the complete curated non-redundant studies and accessed on June 18, 2021. Cohorts were stratified based on genomic status of amplification of *ENPP2*, *MYC*, or wildtype for both genes. Amplification cohorts are tumors displaying amplification of either *ENPP2* or *MYC* in the absence of a co-occurring alteration in the other gene. The ANOVA statistical test with post-hoc analysis was performed where ***$p < 0.0001$. **B** mRNA z-scores of exhaustion markers from TCGA data with samples stratified by high and low *ENPP2* expression representing the top 25% and bottom 25% of *ENPP2* expressing melanoma tumors. Descriptive statistics are as follows for low *ENPP2*: number of values = 91, minimum = −21.76, 25% percentile = −8.777, median = −2.278, 75% percentile = 3.966, maximum = 11.80, range = 33.56, mean = −2.863. Descriptive statistics are as follows for high *ENPP2*: number of

values = 91, minimum = −24.17, 25% percentile = −6.980, median = 2.438, 75% percentile = 9.988, maximum = 17.75, range = 41.92, mean = 1.634. Statistics show the unpaired two-sided Student's t-test analysis was performed where $n = 363$ samples and ***$p < 0.0005$ where $p = 0.0004$. **C, D** tSNE plots of (**C**) *LPAR5* expression in melanoma and immune cells and (**D**) corresponding immune cell populations. tSNE plots were generated using the Single Cell Portal (https://singlecell.broadinstitute.org/single_cell). **E** Two-sided Spearman correlation analysis of *LPAR5* expression and "exhaustion" signature from bulk RNA sequencing on TCGA melanoma tumors where $p < 0.0001$. **F** Relative abundance of LPA in stage IV melanoma responder patients (blue symbols; complete response and partial response where $n = 3$ patients) or non-responders (red symbols; stable disease and progressive disease where $n = 6$ patients) measured both pre- and post-treatment. The unpaired two-sided Student's t-test analysis was performed where *$p < 0.05$ and $p = 0.0313$. Error bars represent standard error of the mean.

Previously, our laboratory has shown *Lpar5*⁻/⁻ CD8 T cells impede local tumor growth better than wildtype CD8 T cells[8,16]. Thus, we first used an orthotopic tumor model to investigate how Lpar5 modulates CD8 T cell exhaustion, however, we did not observe any significant difference in CD8 T cell exhaustion markers in this model (Supplementary Fig. 5). Using our systemic in vivo tumor model, we

investigated additional markers of CD8 T cell exhaustion and these analyses showed that transferred CD45.1⁺ *Lpar5*⁻/⁻ OT-I CD8 T cells isolated from tumors in the lungs expressed reduced amounts of Lag3 and Tox as compared to wildtype transferred CD45.1⁺ OT-I CD8 T cells (Fig. 3K–P). Exhausted CD8 T cells exhibit impaired cytokine production[29–31] so, we also measured interferon γ (IFNγ) and tumor

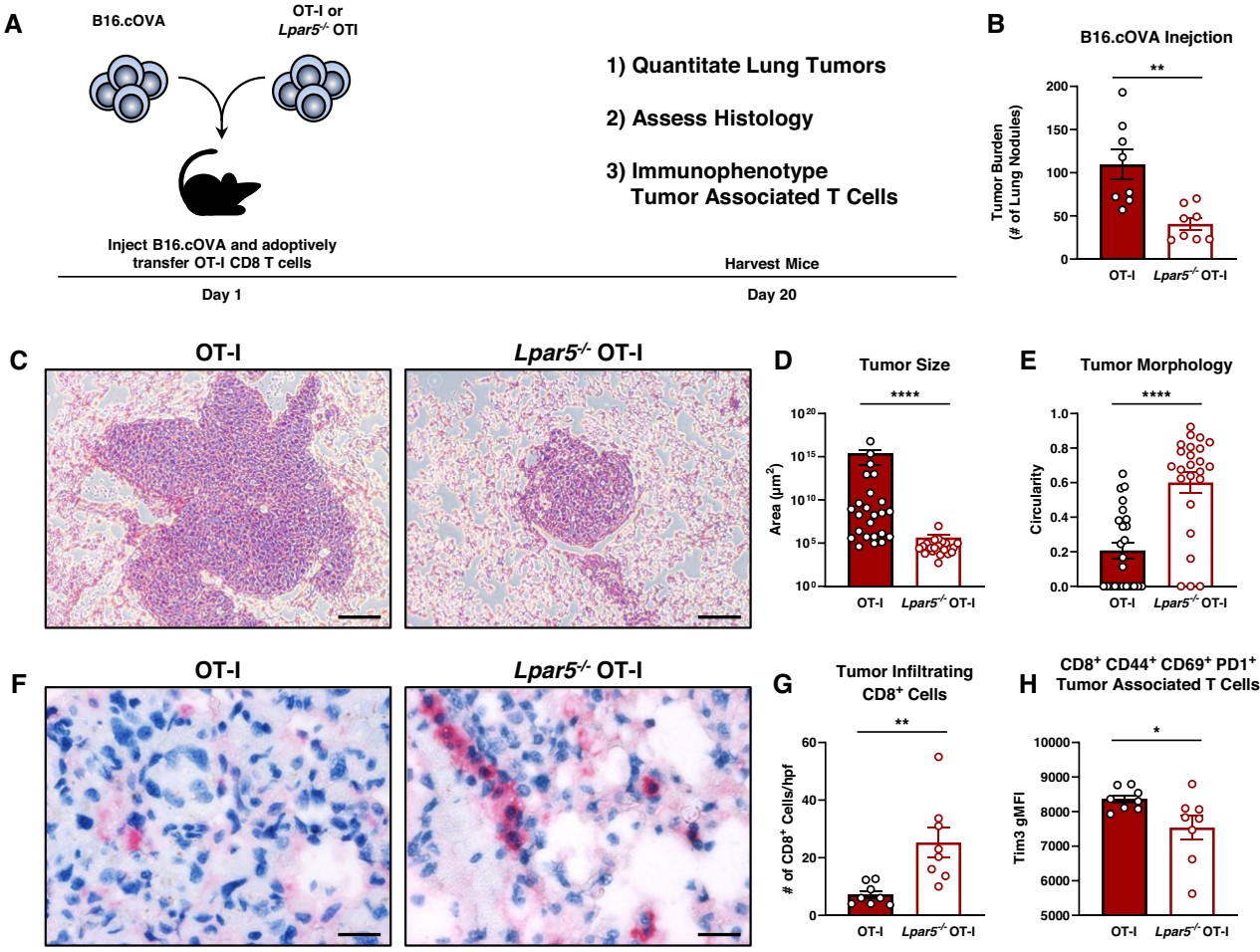

**Fig. 2 | *Lpar5*⁻/⁻ OT-I mice have improved tumor clearance and anti-tumor immunity. A** Schematic showing study design where B16.cOVA tumor cells and OT-I CD8 T cells are co-transferred in a 1:1 ratio into the mice on day 1. Mice were harvested on day 20 and evaluated for tumor burden and flow cytometric and histological analyses were performed on the lung bearing tumors. **B** Quantified tumor burden in the lung after intravascular injection of B16.cOVA cells. Tumor burden is presented as the number of tumor nodules in the lung and *n* = 8 mice per group and *p* = 0.0024. **C** Representative hematoxylin & eosin (H&E) histology images of the B16.cOVA tumor seeded in the lungs. Scale bars represent 100 μm. **D** Tumor area (μm²) quantified from H&E histology of B16.cOVA lung tumors (3 tumors per lung were analyzed and all measurements were averaged and *p* < 0.0001). **E** Circularity analysis of B16.cOVA lung tumors quantified from H&E histology (3 tumors per lung were analyzed and all measurements were averaged and *p* < 0.0001). **F** Representative images of immunohistochemistry for CD8 on lung sections. CD8 positive cells are shown as red to distinguish between melanin and positive stain. Scale bars represent 50 μm. **G** Quantification of intratumoral CD8 positive cells per high powered field (hpf, 400x, averaged values of 3 different tumors per mouse with technical error propagated, *p* = 0.0041). **H** Flow cytometric quantification of Tim3 expression by CD8⁺ CD44⁺ CD69⁺ PD1⁺ tumor-associated T cells expressed as geometric mean fluorescence intensity (gMFI) where *p* = 0.0417. Statistics for this entire figure were performed using the unpaired two-sided Student's *t*-test analysis was performed where \**p* < 0.05, \*\**p* < 0.005, \*\*\**p* < 0.0005, \*\*\*\**p* < 0.0001. Error bars represent standard error of the mean.

necrosis factor α (TNFα) production using our in vivo tumor model and observed that there were modest, albeit non-significant increases in dual IFNγ and TNFα production by transferred CD45.1⁺ *Lpar5*⁻/⁻ OT-I CD8 T cells as compared to wildtype CD45.1⁺ OT-I CD8 T cells (Supplementary Fig. 6). In addition, assessing CD8 T cell cytotoxicity and function as measured by IFNγ⁺ and surface CD107a⁺, we observed a supportive but non-significant trend that *Lpar5*⁻/⁻ OT-I CD8 T cells display increased cytotoxicity as compared to wildtype OT-I CD8 T cells. Altogether, these data provide evidence that Lpar5 signaling on CD8 T cells reprograms phenotypes and to promote an exhaustion-like state both in vitro and in vivo.

**Lpar5 signaling impairs antigen-specific CD8 T cell killing in vivo in a host with a wildtype T cell repertoire**

We next hypothesized that Lpar5-deficient CD8 T cells may have enhanced killing ability in vivo in a wildtype T cell repertoire. To assess this, we performed in vivo killing assays[32] in wild type and *Lpar5*⁻/⁻ mice immunized with N4[33] (Fig. 4A). Immunized wildtype and *Lpar5*⁻/⁻ mice

were subsequently injected with N4-pulsed or unpulsed target cells at a 1:1 ratio (Fig. 4B). As a negative control, HSV1 peptide-pulsed and unpulsed target cells were also injected into immunized mice (Fig. 4C). We confirmed that both wildtype and *Lpar5*⁻/⁻ mice generated similar numbers of ovalbumin-specific (tetramer⁺) CD8 T cells after immunization at day 5 (Fig. 4D, Supplementary Fig. 7). The results from these analyses revealed that one day after transfer of peptide-pulsed target cells, antigen-specific killing was significantly enhanced in Lpar5-deficient mice when compared to wildtype mice (Fig. 4E). Since B cells and macrophages also express *LPAR5* (Fig. 1D), we sought to assess the CD8 T cell-specific contribution to antigen-specific killing in vivo and performed this experiment with an adoptive transfer of wildtype OT-I or *Lpar5*⁻/⁻ OT-I CD8 T cells (Fig. 4F). Using this adoptive transfer model, and consistent with findings in Fig. 4E, we observed that mice transferred with *Lpar5*⁻/⁻ OT-I CD8 T cells exhibit improved antigen-specific killing in vivo as compared to mice transferred with wildtype OT-I CD8 T cells (Fig. 4G–H). Altogether, these results demonstrate Lpar5 signaling acts to negatively modulate antigen-specific CD8 T cell killing in vivo.

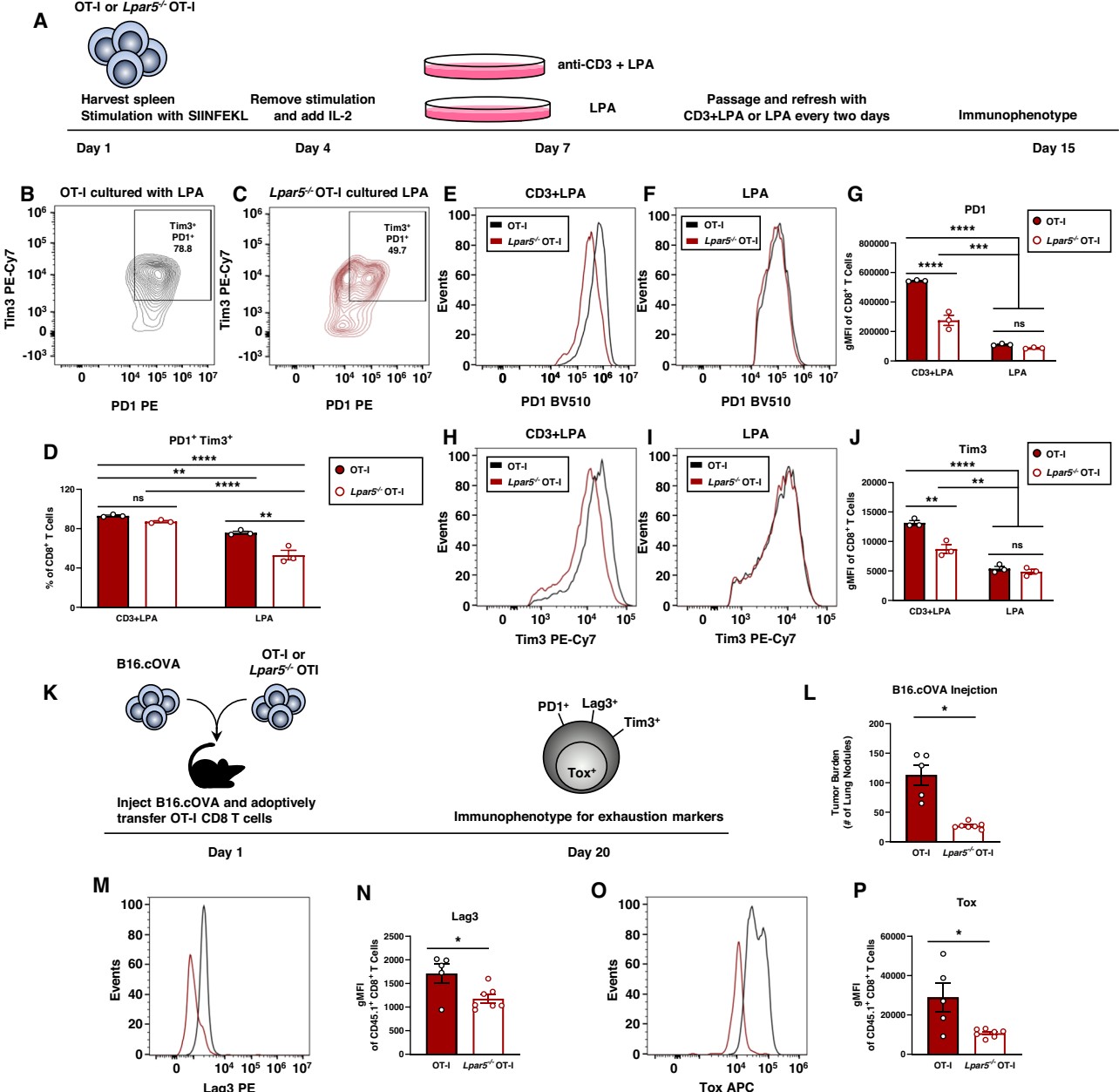

**Fig. 3 | Lpar5 signaling on CD8 T cells promotes phenotypic tolerogenic states through exhaustive-like differentiation. A** Schematic of in vitro chronic stimulation. Effector CD8 T cells are persistently cultured with either anti-CD3 + LPA or LPA alone until Day 15. **B–D** Flow cytometric analysis and quantification of percent PD1+ Tim3+ from the CD8 T cell population. **B, C** Representative contour plots for (**B**) OT-I or (**C**) *Lpar5−/−* OT-I CD8 T cells persistently cultured with LPA. **D** Quantification of percent of CD8+ T cells that are PD1+ Tim3+ (n = 3 mice per group). Exact p-values are as follows, OT-I CD3 + LPA vs OT-I LPA p = 0.0049; OT-I CD3 + LPA vs *Lpar5−/−* OT-I LPA p < 0.0001; *Lpar5−/−* OT-I CD3 + LPA vs *Lpar5−/−* OT-I LPA p < 0.0001; OT-I LPA vs *Lpar5−/−* OT-I LPA p = 0.0012 (**E–G**) Flow cytometric analysis of Tim3 expression on OT-I or *Lpar5−/−* OT-I CD8 T cells cultured in either anti-CD3 + LPA or LPA. **E, F** Representative flow cytometric histograms and (**G**) quantification the geometric mean fluorescence intensity (gMFI) of PD1 (n = 3 mice). Exact p-values are as follow, OT-I CD3 + LPA vs *Lpar5−/−* OT-I CD3 + LPA p < 0.0001; OT-I CD3 + LPA vs OT-I LPA p < 0.0001; OT-I CD3 + LPA vs *Lpar5−/−* OT-I LPA p < 0.0001; *Lpar5−/−* OT-I CD3 + LPA vs OT-I LPA p = 0.0007; *Lpar5−/−* OT-I CD3 + LPA vs *Lpar5−/−* OT-I LPA p = 0.0003. **H–J** Flow cytometric analysis of Tim3 expression on OT-I or *Lpar5−/−* OT-I CD8 T cells cultured in either anti-CD3 + LPA or LPA. **H, I** Representative flow cytometric histograms and (**J**) quantification the geometric mean fluorescence intensity (gMFI) of Tim3 (n = 3 mice). Exact p-values

are as follows, OT-I CD3 + LPA vs *Lpar5−/−* OT-I p = 0.0012; OT-I CD3 + LPA vs OT-I LPA p < 0.0001; OT-I CD3 + LPA vs *Lpar5−/−* OT-I LPA p < 0.0001; *Lpar5−/−* CD3 + LPA vs OT-I LPA p = 0.0044; *Lpar5−/−* OT-I CD3 + LPA vs *Lpar5−/−* OT-I LPA p = 0.0031. **K** Schematic of study design where B16.cOVA tumor cells and OT-I CD8 T cells are co-transferred in a 1:1 ratio into the mice on day 1. Mice were harvested on day 20 and evaluated for tumor burden and flow cytometric analysis for exhaustion markers. **L** Quantified tumor burden in the lung after intravascular injection of B16.cOVA cells. Tumor burden is presented as the number of tumor nodules in the lung where n = 5 mice per OT-I group and n = 7 mice per *Lpar5−/−* OT-I group and p = 0.0124. (M,N) Flow cytometric quantification of Lag3 expression with a (**M**) representative histogram and (**N**) quantification of CD45.1+ CD8+ T cells represented as gMFI where n = 5 mice per OT-I group and n = 7 mice per *Lpar5−/−* OT-I group and p = 0.0216. **O, P** Flow cytometric quantification of Tox expression with a (**O**) representative histogram and (**P**) quantification of CD45.1+ CD8+ T cells represented as gMFI where n = 5 mice per OT-I group and n = 7 mice per *Lpar5−/−* OT-I group and p = 0.0151. Statistics for panels (**D, G, J**) were performed using a Two-way ANOVA with a Tukey's post-hoc analysis where *p < 0.05, **p < 0.005, ***p < 0.0005, ****p < 0.0001. Statistics for panels (**L, N, P**) were performed using the unpaired two-sided Student's t-test analysis where *p < 0.05. Error bars for panels (**D, G, J, L, N, P**) represent standard error of the mean.

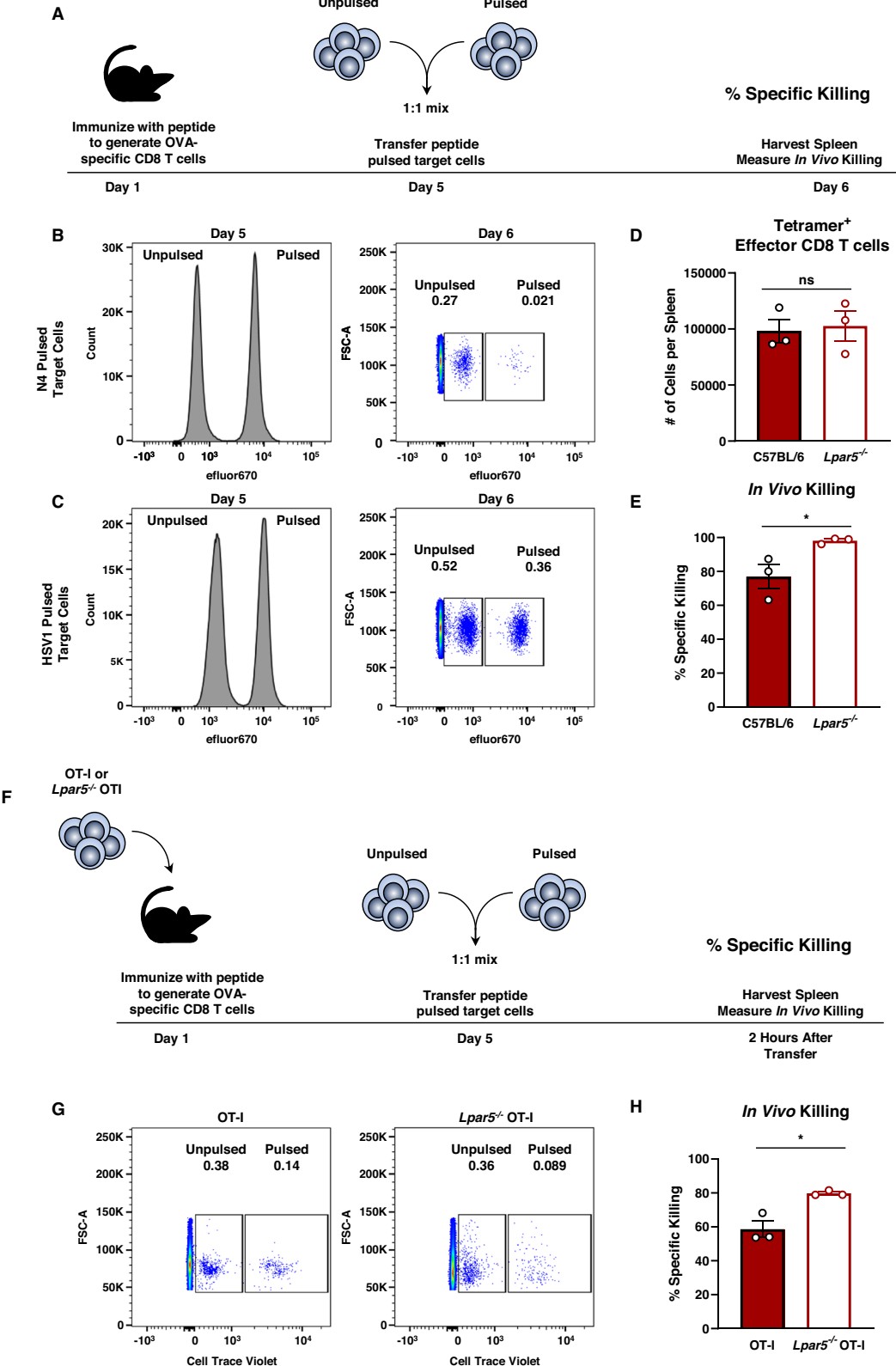

## LPA acts on CD8 T cells to modulate metabolism and production of reactive oxygen species

Both metabolic fitness and antigen-specific killing are critical determinants of CD8 T cell responses during immunotherapy[34] and we hypothesized that in addition to antigen-specific killing, metabolic fitness is also regulated by LPA. Thus, we sought to investigate how LPA signaling modulates effector CD8 T cell metabolism. We first generated effector

OT-I CD8 T cells ex vivo in the presence and absence of LPA. We use the term 'effector CD8 T cells' as CD8+ CD44+ ovalbumin-specific OT-I CD8 T cells generated by stimulation and differentiation during ex vivo cell culture (Supplementary Fig. 8A). We validated that (1) these effector CD8 T cells were homogenously CD8+ CD44+, (2) LPA administration did not skew the effector CD8 T cell population, and (3) LPA treatment did not decrease CD8 T cell viability (Supplementary Fig. 8B–D). We then

**Fig. 4 | Signaling via Lpar5 modulates antigen-specific killing in vivo.**
**A** Schematic of in vivo killing assay. **B, C** Representative flow cytometric histograms and dot plots of target cell input (left) and killing of target cells (right) pulsed with (**B**) N4 ovalbumin peptide or (**C**) HSV1 irrelevant peptide. **D** Frequency of ovalbumin-specific (tetramer⁺) CD8 T cells from the spleens of wildtype C57BL/6 mice and *Lpar5*⁻/⁻ mice immunized with N4 ovalbumin peptide 4 days earlier where $n = 3$ mice per group and $p = 0.8065$. **E** Quantitative analysis of percent specific in vivo killing 5 days after ovalbumin peptide immunization and 1 day after transfer of pulsed target cells from panels A and B where $n = 3$ mice per group and $p = 0.0422$. **F** Schematic of adoptive transfer for in vivo killing assay. **G** Representative flow cytometric dot plots target cell killing pulsed with N4 ovalbumin peptide. **H** Quantitative analysis of percent specific in vivo killing 4 days after ovalbumin peptide immunization and 2 h after transfer of pulsed target cells where $n = 3$ mice per group and $p = 0.0123$. Statistics for this entire figure were performed using the unpaired two-sided Student's *t*-test analysis was performed where *$p < 0.05$. Error bars for panels (**D, E, H**) represent standard error of the mean.

performed mass spectrometry to examine the global metabolic profile and data variance of effector CD8 T cells given media without LPA (RPMI+Glutamine) or with 1 μM LPA for 30 min, 2 h, or 4 h prior (Fig. 5, Supplementary Fig. 8E, F). Metabolites associated with D-glutamine, D-glutamate, and reactive oxygen species (ROS) were found to be enriched in CD8 T cells treated with LPA in unbiased enrichment analyses (Fig. 5A–B). Notably, the levels of metabolites associated with the γ-glutamyl pathway (Fig. 5C), including γ-glutamyl-D-Alanine, 5-oxoproline, and L-glutamate were identified to flux significantly in response to LPA treatment (Fig. 5D–F). The γ-glutamyl cycle serves as a synthesis pathway for regenerating glutathione and specifically, γ-glutamyl-D-Alanine can serve as a metabolite reservoir for glutathione synthesis[35]. Interestingly, we found that glutathione levels did not change with LPA treatment (Fig. 5G), yet fluxes in γ-glutamyl cycle metabolites implicates this pathway as a potential mechanism to preserve glutathione levels in response to LPA treatment. We questioned if LPA regulated the production of ROS in effector CD8 T cells and performed a luciferase assay to directly measure $H_2O_2$ (Fig. 5H, I). These findings demonstrated effector CD8 T cells varied the production of ROS significantly in response to LPA and accumulated ROS with increasing concentrations of LPA. Taken together, these data show LPA rewires CD8 T cell metabolism and modulates ROS levels.

### LPA signaling modulates mobilization of lipids for mitochondrial oxidation in CD8 T cells

Thus far, our data shows that LPA signaling modulates the global metabolic profile of effector CD8 T cells and likely leads to a functional change in CD8 T cell metabolism. To assess metabolic function, we used the Seahorse mitostress test to measure oxygen consumption rate (OCR) and extracellular acidification rate (ECAR) in naïve and OT-I effector CD8 T cells (Fig. 6A–D). We observed LPA treatment of effector CD8 T cells led to an increase in basal respiration, maximal respiratory capacity, and proton leak in addition to a transient increase in ATP-linked production (Fig. 6E–H). Importantly, these findings with transgenic OT-I effector CD8 T cells stimulated in an antigen-specific manner were further validated using wildtype (non-transgenic) naïve and effector CD8 T cells stimulated with anti-CD3/CD28 ex vivo (Supplementary Fig. 8A, G–H).

Since the data show LPA increases ECAR and simultaneously elevates respiratory capacity, we hypothesized that glycolytic products are shuttled out as lactate and endogenous metabolic shunting from non-glycolytic sources was increasing mitochondrial metabolism. We questioned if lipids were consumed for mitochondrial respiration. We measured storage fats in the form of lipid droplets using BODIPY staining and found that LPA treatment rapidly depletes lipid droplets in effector CD8 T cells (Fig. 6I, J). Using etomoxir, we inhibited long chain fatty acid uptake into the mitochondria and reversed the LPA-mediated increase in maximal respiration (Fig. 6K and Supplementary Fig. 9). These findings together show LPA signaling by CD8 T cells shifts metabolism to consume fatty acids for mitochondrial respiration. In summary, LPA signaling changes functional metabolism resulting in increased fatty acid oxidation and proton leak in CD8 T cells.

### Metabolic efficiency is determined by Lpar5 in CD8 T cells

Since we observed improved tumor immunity by Lpar5-deficient CD8 T cells, we sought to determine the Lpar5 receptor contribution on CD8 T cell metabolism. We performed the Seahorse mitostress test on *Lpar5*⁻/⁻ OT-I effector CD8 T cells in absence of LPA and found similar basal respiration but increased maximal respiratory capacity compared to wildtype OT-I effector CD8 T cells (Fig. 7A). Treatment with an Lpar5 antagonist (TC LPA5 4) did not affect basal respiration but did abrogate the LPA-mediated increase in maximal respiratory capacity in wildtype OT-I effector CD8 T cells (Fig. 7B). Importantly, these data show that the *Lpar5*-deficient CD8 T cells have greater reserve to increase their respiratory capacity relative to wildtype effector CD8 T cells (Fig. 7C). Capacity calculations confirmed Lpar5 modulates maximal respiration and proton leak but not basal respiration in effector CD8 T cells (Fig. 7D–G). Interestingly, increased proton leak from LPA treatment can be rescued with receptor antagonism and Lpar5 deficiency further decreases proton leak. We further confirmed these results by treating *Lpar5*⁻/⁻ OT-I effector CD8 T cells with LPA and found an increase basal but not maximal respiration or proton leak (Fig. 7H–M). In contrast to transient ATP production observed with Lpar5-sufficient T cells (Fig. 6B), ATP production was sustained at high levels in *Lpar5*⁻/⁻ OT-I effector CD8 T cells in response to LPA. Since total amount of mitochondria could affect maximal respiratory capacity, we performed MitoTracker staining as a semi-quantitative measure of mitochondrial mass. We observed a subtle and transient decrease in MitoTracker in both the wildtype OT-I and *Lpar5*⁻/⁻ OT-I effector CD8 T cells treated with 1 μM LPA (Supplementary Fig. 10A–D). However, we did not observe a significant difference in total mitochondrial mass between wildtype OT-I and *Lpar5*⁻/⁻ OT-I effector CD8 T cells in the absence of LPA (Supplementary Fig. 10E–F). In sum, these data reveal *Lpar5*⁻/⁻ OT-I effector CD8 T cells have more efficient and flexible metabolism in response to LPA treatment.

## Discussion

In this study, we found that LPA signaling is a tolerogenic mechanism to regulate CD8 T cell metabolism and impair anti-tumor immunity. We demonstrate metabolic efficiency, performance, and antigen-specific CD8 T cell killing are modulated by Lpar5. LPA signaling by effector CD8 T cells acutely promotes lipolysis, mitochondrial fatty acid uptake, and increased proton leak. Further, we identify plasma LPA levels as a potential predictor of response to CD8 T cell mediated therapies in stage IV melanoma patients. These data not only add to the existing body of evidence that LPA and ATX play an important role in tumor progression, but also provide convincing evidence that lipid signaling could be exploited as an approach to (1) prevent and/or reinvigorate dysfunctional CD8 T cells, (2) promote effective endogenous anti-tumor immunity, and (3) improve clinical responses to immunotherapy.

We determined that Lpar5 signaling in effector CD8 T cells modulates maximal respiratory capacity but not basal respiration. Effector CD8 T cells predominantly express Lpar2, Lpar5, and Lpar6[8]. This would suggest that LPA signaling through either Lpar2 or Lpar6 is responsible for increases in basal metabolism in response to LPA but requires further study. Maximal respiratory capacity is an indicator of CD8 T cell adaptability to energy demands and is important to consider in the context of other metabolic capacities. While we found that LPA increased maximal respiratory capacity, it also increased proton leak and did not lead to sustained ATP production. Taken together, these data suggest that LPA-mediated changes to CD8 T cell

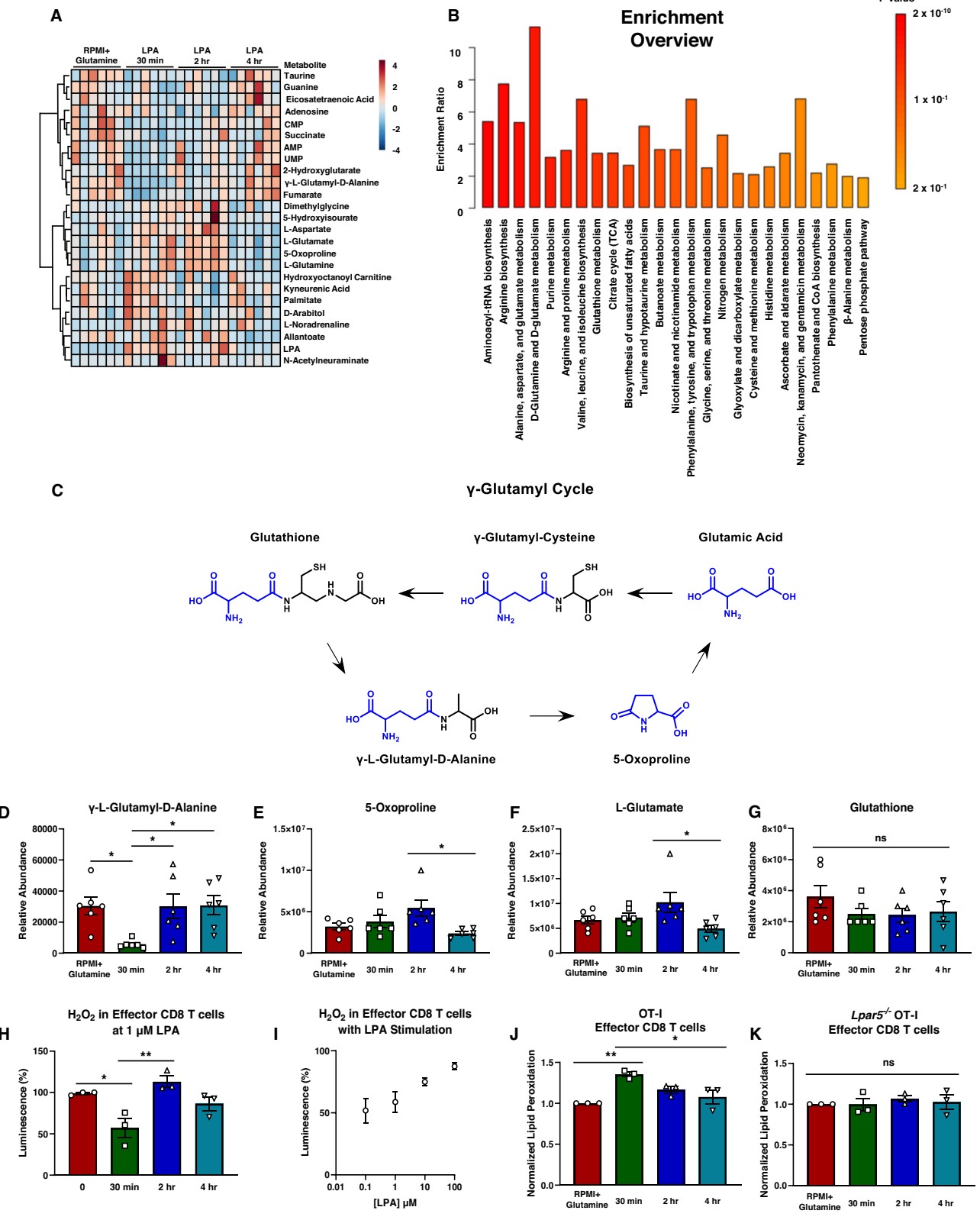

metabolism result in inefficient bioenergetics. Lpar5-deficient CD8 T cells have increased maximal respiration, sustained ATP production, and low levels of proton leak indicative of more efficient metabolism. Metabolic performance and sustained ATP are essential for interferon production, NLRP3 inflammasome activation, and recall capacity[13,15,36]. Thus, Lpar5 inhibition of CD8 T cell activation and function likely serves as a metabolically regulated immune checkpoint.

There is an energetic cost associated with immune synapse formation and the release of perforin, granzyme, and cytokines at the immunological synapse which are all required for efficient tumor cell killing. Data from our laboratory has shown that LPA signaling interferes with perforin and cytokine release at the immunological synapse, in part by co-opting of the actin and microtubule cytoskeleton[8,37]. Previous reports have shown that mitochondria localize to the T cell

**Fig. 5 | Lysophosphatidic acid rewires CD8 T cell metabolism and modulates reactive oxygen species. A** Mass spectrometry showing global metabolomic data on effector CD8 T cells given media without LPA (RPMI+Glutamine) or treated with 1 μM LPA for 30 min, 2 h, or 4 h prior to sample collection where $n = 6$ mice per group and shows Euclidean clustering analysis. **B** Metabolite set enrichment analysis (MSEA) performed on raw data with KEGG analysis to determine enriched metabolic pathways. **C** Metabolic pathway of γ-glutamyl cycle where blue represents the recycled atoms from γ-L-glutamyl-D-alanine to synthesize glutathione. **D–G** Relative intracellular abundances of (**D**) γ-L-glutamyl-D-alanine and exact $p$-values are as follows, RPMI+Glutamine vs 30 min LPA $p = 0.0313$; 30 min LPA vs 2 h LPA $p = 0.0347$; 30 min LPA vs 4 h LPA $p = 0.0290$, (**E**) 5-oxoproline and exact $p$-values are as follows, 2 h LPA vs 4 h LPA $p = 0.0143$, (**F**) L-glutamate and exact $p$-values are as follows, 2 h LPA vs 4 h LPA $p = 0.0262$, and (**G**) glutathione with $n = 6$ mice per group. **H, I** Direct measurements of $H_2O_2$ in effector CD8 T cells after LPA

treatment (**H**) at 1 μM for 30 min, 2 h, or 4 h or (**I**) at varying concentrations of LPA after 15 min of LPA treatment with $n = 3$ mice per group. Data measuring reactive oxygen species are normalized to cells cultured in the absence of LPA. **J, K** Measurements of lipid peroxidation in effector CD8 T cells after LPA treatment at 1 μM for 30 min, 2 h, or 4 h in (**J**) OT-I effector CD8 T cells and (**K**) $Lpar5^{-/-}$ OT-I CD8 T cells with $n = 3$ mice per group. For (**H–K**) samples were measured in technical triplicates and error was propagated to biological replicate error where $n = 3$ mice per group performed in 3 independent experiments. Exact $p$-values for (**H**) are as follows, 0 vs 30 min LPA $p = 0.0239$; 30 min LPA vs 2 h LPA $p = 0.0045$. Exact $p$-values for (**J**) are as follows, RPMI+Glutamine vs 30 min LPA $p = 0.0039$; 30 min LPA vs 4 h LPA $p = 0.0166$. Statistics for this entire figure were performed using an ANOVA statistical test with Tukey's post-hoc analysis was performed where $*p < 0.05$ and $**p < 0.005$.

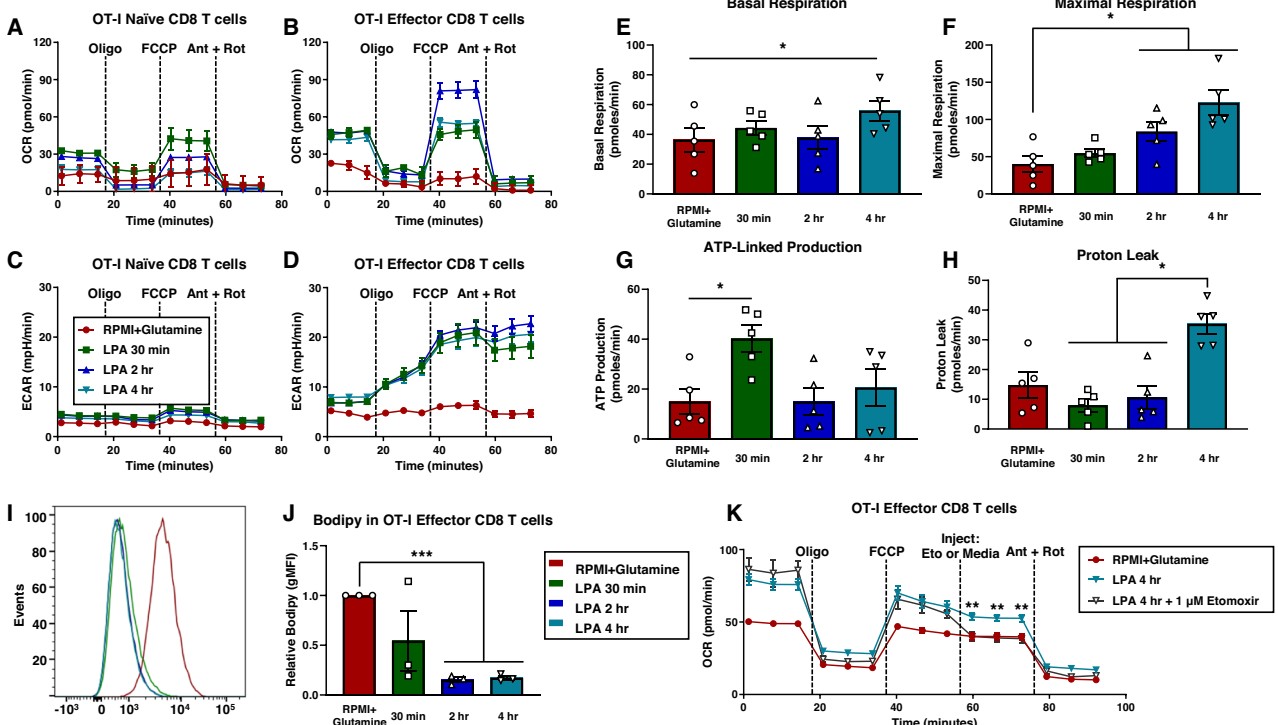

**Fig. 6 | Lysophosphatidic acid shifts metabolism to consume fatty acids for oxidation. A–D** Oxygen consumption rate (OCR) and extracellular acidification rate (ECAR) by both naïve and effector CD8 T cells given media without LPA (RPMI+Glutamine) or treated with 1 μM LPA for 30 min, 2 h, or 4 h prior to starting the Seahorse metabolic flux assay. Assay was performed with injections of oligomycin (oligo), (4-(trifluoromethoxy) phenyl) carbonohydrazonoyl dicyanide (FCCP), antimycin A (ant), and rotenone (rot) at 18-min intervals in media supplemented with 25 mM glucose. Data are representative and show $n = 6$ technical replicates. **E–H** Capacity calculations from Seahorse metabolic flux assay showing basal respiration, maximal respiration, ATP-linked production, and proton leak. Data show $n = 5$ independent experiments with technical replicate error propagated into biological replicate error. Exact $p$-values for (**E**) are as follows, RPMI+Glutamine vs 4 h LPA $p = 0.0149$. Exact $p$-values for (**F**) are as follows, RPMI+Glutamine vs 2 h LPA $p = 0.0455$; RPMI+Glutamine vs 4 h LPA $p = 0.0489$. Exact $p$-values for (**G**) are as follows, RPMI+Glutamine vs 30 min LPA $p = 0.0491$. Exact $p$-values for panel (**H**) are as follows, 30 min LPA vs 4 h LPA $p = 0.0129$; 2 h LPA vs

4 h LPA $p = 0.0347$. **I, J** Flow cytometric analysis of BODIPY in effector CD8 T cells given media without LPA (RPMI+Glutamine) or treated with 1 μM LPA for 30 min, 2 h, or 4 h. **I** Shows representative histogram and (**J**) shows quantitative analysis of normalized geometric mean fluorescence intensity (gMFI) across $n = 3$ independent experiments with 3 mice per group. Exact $p$-values are as follows, RPMI+Glutamine vs 2 h LPA $p = 0.0002$; RPMI+Glutamine vs 4 h LPA $p = 0.0002$. **K** Seahorse metabolic flux analysis performed with acute injection of etomoxir to a final concentration of 1 μM. Effector CD8 T cells were cultured in normal media (RPMI+Glutamine) or 1 μM LPA for 4 h prior to starting the assay. $n = 6$ technical replicates. Exact $p$-values are as follows, at $t = 60$ min RPMI+Glutamine vs 4 h LPA $p = 0.0049$; 4 h LPA vs 4 h LPA + Etomoxir $p = 0.0044$, at $t = 66$ min RPMI+Glutamine vs 4 h LPA $p = 0.0045$; 4 h LPA vs 4 h LPA + Etomoxir $p = 0.0042$, at $t = 72$ min RPMI+Glutamine vs 4 h LPA $p = 0.0030$; 4 h LPA vs 4 h LPA + Etomoxir $p = 0.0030$. Statistics for (**E–K**) were performed using an ANOVA statistical test with a Tukey's post-hoc analysis was performed where $*p < 0.05$, $**p < 0.005$, and $***p < 0.0005$. Error bars for panels (**A–H, J, K**) represent standard error of the mean.

immunological synapse[38,39]. In this report, we found increased maximal respiratory capacity in $Lpar5^{-/-}$ OT-I CD8 T cells which reflects a greater ability to adapt to energetic demands. As such, LPA signaling likely plays a role in modulating the energetic burst required to release perforin and granzyme at the immunological synapse. Our data suggest that LPA signaling through Lpar5, which would be heightened in

individuals with certain cancers, results in dysfunctional metabolism unable to sustain the energy required for optimal antigen-specific killing.

Our data revealed that LPA induced a rapid depletion of neutral lipids in effector CD8 T cells. Triglycerides from neutral lipids are catabolized to free fatty acids which are shuttled into the mitochondria

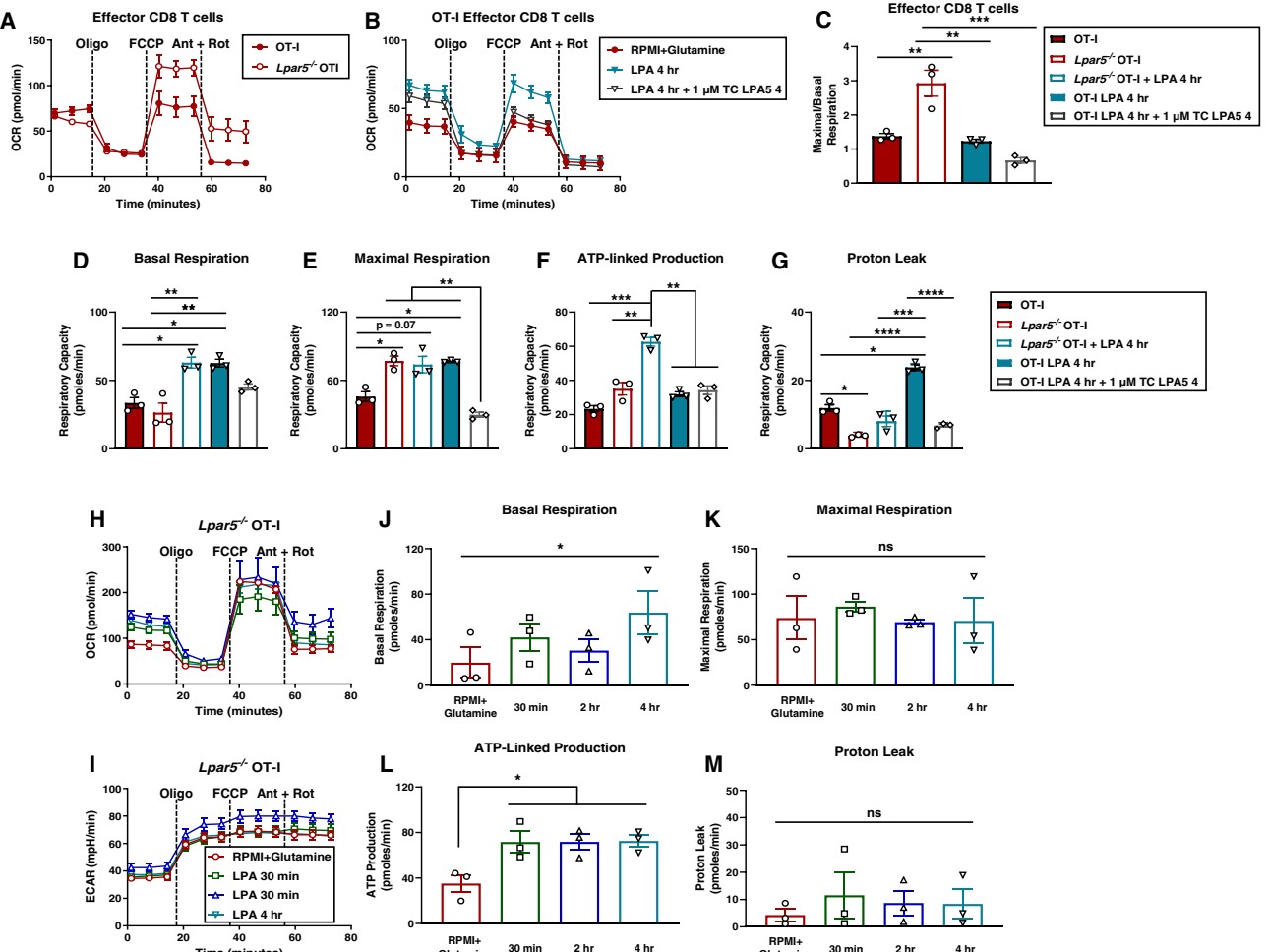

**Fig. 7 | Lysophosphatidic acid receptor 5 modulates metabolic adaptability and efficiency in effector CD8 T cells. A, B** Seahorse metabolic flux assay performed on effector CD8 T cells from (**A**) an OT-I mouse or *Lpar5⁻/⁻* OT-I mouse in the absence of LPA treatment and (**B**) given normal media without LPA (RPMI+Glutamine), treated with 1 μM LPA for 4 h prior to starting the assay, or co-treatment of 1 μM LPA and the LPA receptor antagonist (TC LPA5 4 at 1 μM) for 4 h prior to starting the assay. Data are representative and show *n* = 6 technical replicates. **C–G** Capacity calculations from Seahorse metabolic flux assay showing (**C**) ratio of maximal respiratory capacity / basal respiratory capacity where exact *p*-values are as follows, OT-I vs *Lpar5⁻/⁻* OT-I *p* = 0.0026; *Lpar5⁻/⁻* OT-I vs OT-I 4 h LPA *p* = 0.0015; *Lpar5⁻/⁻* OT-I vs OT-I 4 h LPA + TC LPA5 4 *p* = 0.0002, **D** basal respiration where exact *p*-values are as follows, OT-I vs *Lpar5⁻/⁻* OT-I + 4 h LPA *p* = 0.048; OT-I vs OT-I 4 h LPA *p* = 0.0052; *Lpar5⁻/⁻* OT-I vs *Lpar5⁻/⁻* OT-I + 4 h LPA *p* = 0.0010; *Lpar5⁻/⁻* OT-I vs OT-I 4 h LPA *p* = 0.0010, (**E**) maximal respiration where exact *p*-values are as follows, OT-I vs *Lpar5⁻/⁻* OT-I *p* = 0.0052; OT-I vs *Lpar5⁻/⁻* OT-I + 4 h LPA *p* = 0.07; OT-I vs OT-I 4 h LPA *p* = 0.0055; *Lpar5⁻/⁻* OT-I + 4 h LPA vs OT-I 4 h LPA *p* = 0.0049; *Lpar5⁻/⁻* OT-I + 4 h LPA vs OT-I + 4 h LPA + TC LPA5 4 *p* = 0.0042; OT-I 4 h LPA vs OT-I 4 h LPA + TC LPA5 4 *p* = 0.0049, (**F**) ATP-linked production where exact *p*-values are as follows, OT-I vs *Lpar5⁻/⁻* OT-I 4 h LPA *p* = 0.0001; *Lpar5⁻/⁻* OT-I vs *Lpar5⁻/⁻* OT-I + 4 h LPA *p* = 0.0041;

*Lpar5⁻/⁻* OT-I + 4 h LPA vs OT-I + 4 h LPA *p* = 0.0050; *Lpar5⁻/⁻* OT-I + 4 h LPA vs OT-I + 4 h LPA + TC LPA5 4 *p* = 0.0050, and (**G**) proton leak where exact *p*-values are as follows, OT-I vs *Lpar5⁻/⁻* OT-I *p* = 0.0080; OT-I vs OT-I + 4 h LPA *p* = 0.0050; *Lpar5⁻/⁻* OT-I vs OT-I + 4 h LPA *p* < 0.0001; *Lpar5⁻/⁻* OT-I + 4 h LPA vs OT-I + 4 h LPA *p* = 0.0001; OT-I + 4 h LPA vs OT-I + 4 h LPA + TC LPA5 4 *p* < 0.0001. Data show *n* = 3 independent experiments with technical replicate error propagated into biological replicate error. **H, I** Seahorse metabolic flux assay on effector CD8 T cells from a *Lpar5⁻/⁻* OT-I mouse given media without LPA (RPMI+Glutamine) or treated with 1 μM LPA for 30 min, 2 h, or 4 h prior to starting the assay. Data are representative and show *n* = 6 technical replicates. **J–M** Capacity calculations from Seahorse metabolic flux assay showing basal respiration, maximal respiration, ATP-linked production, and proton leak. Data show *n* = 3 independent experiments with technical replicate error propagated into biological replicate error. Exact *p*-values for (**J**) are as follows, RPMI+Glutamine vs 4 h LPA *p* = 0.0328. Exact *p*-values for (**L**) are as follows, RPMI+Glutamine vs 30 min LPA *p* = 0.0495; RPMI+Glutamine vs 2 h *p* = 0.0485; RMPI+Glutamine vs 4 h LPA *p* = 0.0215. Statistics for this entire figure were performed using an ANOVA statistical test with a Tukey's post-hoc analysis was performed where *p* < 0.05, **p* < 0.005, ***p* < 0.0005, and ****p* < 0.0001. Error bars for panels (A-M) represent standard error of the mean.

for oxidative consumption[40]. However, if triglycerides are broken down from lipid droplets and not used for energetic consumption, then these free fatty acids can become lipotoxic in the cytosol. Previous groups have reported that lipid droplets may serve a protective role by buffering cellular amounts of toxic lipids that cause oxidative stress and lipotoxicity[41,42]. Our results show LPA modulates oxidative stress in CD8 T cells via a metabolic mechanism. Specifically, we observed LPA signaling results in a transient flux of $H_2O_2$ while at the same time a corresponding increase in lipid peroxidation (Fig. 5H, J). We did not observe an increase in lipid peroxidation in *Lpar5⁻/⁻* OT-I effector CD8 T cells (Fig. 5K). Considered together, our

data shows evidence that oxidative damage and lipotoxicity is a consequence of Lpar5 signaling. Yet, we also observed that both wildtype OT-I and *Lpar5⁻/⁻* OT-I effector CD8 T cells treated with LPA exhibit a transient decrease in mitochondrial mass (Supplementary Fig. 10A–D). This transient decrease is subtle, and the exact biological significance of this modulation remains unclear. We also found fluxes in hydroxyglutarate, an oncometabolite previously implicated in tumor metabolism, although its exact role and relevance in CD8 T cell biology remains controversial[43,44]. Kynurenic acid levels were also observed to change with LPA treatment. Kynurenic acid is produced in response to ROS and is associated with neuroprotection[45]. Considering the

metabolic data in aggregate, there are strong implications for LPA modulating effector CD8 T cell production of ROS and subsequent cytotoxic function. Interestingly, previous findings show that mitochondrial stress and ROS drive exhaustion and dysfunction in Tim3[+] PD1[+] CD8 T cells[11]. Considered together with our in vitro and in vivo findings assessing Tim3 and PD1 expression on wildtype OT-I and *Lpar5*[−/−] OT-I CD8 T cells, we speculate that Lpar5-deficient CD8 T cells may have less ROS and mitochondrial stress which could explain our observation of decreased Tim3 and PD1 expression on Lpar5-deficient CD8 T cells. However, future studies are required to understand how LPAR5 and ATX signaling on CD8 T cells may affect ROS accumulation in the long-term.

In sum, our data shows Lpar5 signaling fluxes ROS, results in lipid peroxidation, and increases proton leak. While metabolic state and CD8 T cell exhaustion are highly associated, it is unclear whether the metabolic state induced by LPA alone can drive CD8 T cell exhaustion. It could be possible that LPA drives generalized oxidative damage and targeting metabolism could be a future avenue for investigation. Notably, our current understanding of the mechanisms regulating lipotoxicity, oxidative stress, and proton leak are actively being investigated and evolving[46–49]. A recent publication has challenged our fundamental and dogmatic understanding of proton uncoupling[46]. Importantly, metabolic pathways that result in proton uncoupling, leak, and oxidative damage have been reported to be a key fate-determining mechanism in T cells[11,13,50,51]. Thus, our data contribute to an emerging and important field in immunometabolism which highlight the need for future studies to elucidate the specific mechanisms of how oxidative damage and proton leak are regulated in CD8 T cells.

Notably, *ENPP2*, and its LPA product, are constitutively expressed by certain cells in the body and low levels of LPA in certain tissues is physiologic[52]. The normal physiologic role that LPA plays on CD8 T cells may be related to modulating CD8 T cell surveillance via an unconventional mechanism of peripheral tolerance. LPA levels increase to pathologic levels in a number of cancers and chronic infections[17]. Thus, the resultant LPA signaling may serve as another mechanism to avoid immune destruction and further promote tumor development and progression. Previously, our laboratory has reported that LPA signaling through Lpar5 on B cells and T cells impairs intracellular calcium signaling downstream of the T cell and B cell receptors[16,53]. Interestingly, macrophages and NK cells also express LPAR5 (Fig. 1C–D), yet the exact role and function of LPAR5 on myeloid and other lymphoid lineage cell types remains poorly defined. However, continued investigation of LPAR signaling in CD8 T cells will be instrumental in understanding its relevance and role as an immune checkpoint. We also find that elevated levels of systemically circulating LPA predict responses to immunotherapy (Fig. 1F). While the cohort of patients examined is small, we did observe that plasma LPA levels predicted response to single agent nivolumab (Source Data). There was one other patient in the cohort with low levels of LPA who did not respond to combination ipilimumab/nivolumab (Patient #9). However, after following-up on the clinical data, we found that this patient was later treated with a bispecific anti-PD-1/ICOS antibody and is currently disease free. These findings warrant further investigation into how LPAR5 functions as an immune checkpoint and may affect responses to immune checkpoint blockade with anti-CTLA4, anti-PD-1, and combination anti-CTLA4/anti-PD1 therapy.

We propose LPA signaling reprograms adaptive CD8 T cell immunity and immunosurveillance. The findings presented here identify LPA signaling as a mechanism to regulate metabolic reprogramming and differentiation of dysfunctional CD8 T cells. LPA-induced metabolism is potentially an important determinant of T cell fate and generation of effector versus exhausted-like CD8 T cells. Accordingly, LPA likely serves as a mechanism of tolerance that is exploited by cancer. LPA can result in the stimulation of growth and migration of cancer cells[17,54–56]. Yet, the mechanisms underlying this are not entirely clear despite multiple prior studies in prostate, ovarian, breast, and other cancers[5,56–58]. Recent years have seen significant interest and success in treating various cancers with immunotherapy, primarily through enhancing T cell tumor killing. Our study evaluates how extracellular LPA acts on CD8 T cells to modulate efficiency of CD8 T cell metabolism and the energetic burst required for antigen-specific CD8 T cell killing. Altogether, we establish LPA signaling through Lpar5 as a potential CD8 T cell directed therapy to improve endogenous anti-tumor immune responses. While most of the clinical studies done here were in malignant melanoma, we anticipate a similar mechanism may be observed with other cancers that are currently treated with immunotherapy. Future studies should investigate how LPA, ATX, and LPAR5 signaling modules CD8 T cell phenotypes in these cancer types. Altogether, lipid signaling is a promising and targetable approach to reinvigorate CD8 T cells and improve anti-tumor immunity.

## Methods
All human studies were conducted under approval from the Colorado Institutional Review Board (IRB# 05-0309) with patient consent. All mouse studies were performed in accordance with the regulations of the Institutional Animal Care and Use Committee.

### Melanoma patient samples
Blood samples from melanoma patients were collected from the University of Colorado Health Hospital and details of the collection are described in Supplementary Information.

### Mice
C57BL/6 mice were obtained from Jackson Laboratory (Stock Number 000664). OT-I mice[27] (CD45.1 and CD45.2) expressing Vβ5Vα2 T cell receptor were a gift from Dr. Ross Kedl (University of Colorado Anschutz School of Medicine). *Lpar5*[−/−] and *Lpar5*[−/−] OT-I mice (CD45.1 expressing)[16] were genotyped, bred, and maintained at University of Colorado Anschutz School of Medicine. All mice were on normal diets. Experiments were conducted in both CD45.1 and CD45.2 backgrounds (Supplementary Fig. 11). Expression of Lpar5 was assessed using quantitative real-time PCR. Experiments were performed with both male and female mice at 7–12 weeks of age. All mice were housed under pathogen-free conditions and maintained in accordance with the regulations of the Institutional Animal Care and Use Committee.

### Mass spectrometry, metabolomics, and lipidomics of human and mouse samples
CD8 T cells were collected from ex vivo cultures for global water-soluble metabolomics. Two million cells were pelleted after being cultured with no LPA (RPMI+Glutamine) or LPA for 30 min, 2 h, or 4 h prior to collection. Samples were stored at −80 °C prior to analysis. Metabolites were analyzed by liquid chromatograph/tandem mass spectrometry as previously described[59]. Briefly, water soluble metabolites were extracted at 4 °C in 5:3:2 MeOH:MeCN:water (v/v/v) and the resulting supernatant was analyzed on Thermo Vanquish UHPLC coupled to a Thermo Q Exactive mass spectrometer. All data is provided in Source Data. Lipid metabolites from human plasma samples were analyzed as similarly described[59]. In brief, lipids were extracted at 4 °C in methanol and the resulting supernatant was analyzed on the Thermo Vanquish UHPLC coupled to a Thermo Q Exactive mass spectrometer. Data analysis was performed using MetaboAnalyst and the enrichment analysis was performed to integrate comparisons across all groups.

### Ex vivo stimulation and cell culture of CD8 T cells
Antigen-specific OT-I splenocytes were isolated and homogenized into a single cell suspension. Red blood cells were lysed with 0.83%

NH$_4$Cl-Tris Buffer with 1 ml for 5 min at room temperature. OT-I CD8 T cells were pulsed with SIINFEKL (N4) at 2 µg/ml and incubated for 3 days at 37 °C. SIINFEKL peptide was replaced with fresh media containing IL-2 (200-02, Peprotech, 1000 units/ml). Cells were cultured for an additional 3 days. CD8 T cells were isolated using a Ficoll gradient. Effector CD8 T cells were defined as CD8$^+$ CD44$^+$ cells which were ~95% of the cell population. CD8 T cell cultures were identified and validated using antibodies against CD8 BV421 (53–6.7, Biolegend), CD44 (103018, Biolegend), Vβ5 (MR9-4, Biolegend), and Vα2 (B20.1, Biolegend). For chronic stimulation assays, CD8 T cells were passaged onto plates coated with or without In Vivo Ready Anti-Mouse CD3e (145-2C11, Tonbo Biosciences, 40-0031-U100) and maintained in IL-2.

### Measuring reactive oxygen species and lipid peroxidation
H$_2$O$_2$ was measured using the ROS-Glo H$_2$O$_2$ Assay (Promega). Effector CD8 T cells were plated at 200,000 cells per well and given media without LPA (RPMI+Glutamine) or treated with LPA for 15 min, 30 min, 2 h, or 4 h. Lipid peroxidation was measured using a Thiobarbituric Acid Reactive Substances (TBARS) assay (Cayman Chemical). A standard colorimetric curve was generated each time the assay was performed. Luminescence or colorimetric changes were read on the Synergy 2 plate reader (BioTek) on the same day the cells were plated. Technical triplicates were analyzed for each condition and normalized to no LPA treatment controls. For LPA concentration curve reading, the normalization was set to zero. Biological triplicates were performed, and technical error was propagated.

### Flow cytometry
All antibodies were purchased from Biolegend and each antibody and associated panel is listed in Source Data. Additional details on flow cytometry procedures is provided in Supplementary Information.

### LPA preparation
Lyophilized 18:1 LPA (1-oleoyl-2-hydroxy-sn-glycero-3-phosphate, Avanti Polar Lipids) was aliquoted and stored at −20 °C until use. Aliquots were diluted to 1 mM in RPMI medium supplemented with glutamine. The reconstituted LPA was sonicated for 30 min prior to use.

### Metabolic flux assay
CD8 T cells were plated at 200,000 cells per well in Seahorse media and cultured in a non-CO$_2$ incubator. The CD8 T cells were plated and analyzed on the same day. Oligomycin (75351, Sigma-Aldrich, St. Louis, MO, USA), FCCP ((4-(trifluoromethoxy) phenyl) carbonohydrazonoyl dicyanide, C2920, Sigma), and antimycin A (A8674, Sigma-Aldrich) + rotenone (R8875, Sigma-Aldrich) were used at final concentrations of 2.5 µM, 2.0 µM, 0.5 µM, and 0.5 µM respectively. Drugs were loaded into the cartridge and the cartridge was run on Seahorse XFe96 Analyzer with 96-well plates (Agilent Technologies, Santa Clara, CA, USA). Viability was assessed at the time of Seahorse using flow cytometric analysis. Additional details on this protocol are provided in Supplementary Information.

### In vivo killing assay
In vivo killing assays were performed as previously published[8]. C57BL/6 mice or *Lpar5*$^{-/-}$ mice were immunized as previously described[33] by intravenous injection with 40 µg of anti-CD40, 40 µg of polyinosic-polycytidylic acid (pI:C), 150 µg N4, and PBS. Prior to use, pI:C was incubated at 56 °C for 30 min. Anti-CD40 and pI:C were kindly gifted from Dr. Ross Kedl. For adoptive transfer experiments, 10,000 OT-I or *Lpar5*$^{-/-}$ OT-I CD8 T cells were intravenously injected on the same the day the mice were immunized. Four days post-immunization, mice were intravenously injected with target cells. Target cells were prepared the day of injection from C57BL/6 splenocytes. These target cells were either pulsed or unpulsed with 2 µg/ml of N4 (SIINFEKL) or an irrelevant HSV1 peptide (SSIEFARL). The unpulsed cells were stained with 0.5 µM eFluor 670 or Cell Trace Violet and the pulsed target cells were stained with 5 µM eFluor 670 (65-0840-85, eBioscience). The pulsed and unpulsed cells were then mixed in a 1:1 ratio. A small portion of this mix was evaluated flow cytometrically to determine initial input ratio of target cells. A total of 10,000,000 cells were intravenously injected into the vaccinated mice (5,000,000 peptide pulsed cells and 5,000,000 non-peptide pulsed cells). Spleens from immunized mice were then harvested the next day (-20 h or 2 h after injection), homogenized, lysed for red blood cells, washed, resuspended in FACS buffer, and run on the LSRII flow cytometer.

### In vivo tumor models
The B16 mouse melanoma cell line expressing OVA (B16.cOVA) was kindly provided by Dr. Ross Kedl and mycoplasma tested and STR profiled. B16.cOVA cells were maintained and cultured in complete D-10 media (DMEM + 10% FBS + 1% penicillin-streptomycin). Cells were passaged <10 times prior to use. For systemic tumor models, one million B16.cOVA melanoma cells were intravenously injected into C57BL/6 recipient mice (*n* = 5–8 mice per group). On the same day, OT-I CD8 T cells were isolated from either wildtype or *Lpar5*$^{-/-}$ mice using a magnetic bead isolation protocol (130-104-075, Miltenyi Biotec). One million CD8 T cells were adoptively transferred into C57BL/6 recipient mice by intravenous tail vein injection. Mice were sacrificed 20 days later. Lungs were harvested and tumors were quantified using a dissecting microscope as previously published[60]. Additional experimental details on tumor processing are provided in Supplementary Information.

### Histology and microscopy
FFPE blocks and slides were prepared according to standard protocols. Specific protocol details are provided in the Supplementary Information.

### Statistics and reproducibility
Experiments were completed in at least biological triplicate. Results were expressed as mean ± standard error of the mean. Student's *t*-tests, nonparametric analyses, and ANOVA were used for comparisons. Statistical power was considered to determine mouse cohort sizes so that meaningful comparisons can be made between groups.

### Reporting summary
Further information on research design is available in the Nature Portfolio Reporting Summary linked to this article.

## Data availability
The data generated in this study are available within the article and its supplementary information/source data files. Source data are provided with this paper.

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

## Acknowledgements

We wish to thank the patients and their families for donating blood and tissue to the International Melanoma Biorepository and Research Laboratory (IMBRL) at the University of Colorado Cancer Center. We want to thank the Melanoma Scientific Advisory Board at the University of Colorado for their support. We also wish to acknowledge the support provided by the Hertz Foundation, the Amy Davis Foundation, the Moore Family Foundation, and the Heidi Horner Foundation. This study was funded by in part by grants from the National Institutes of Health to R.M.T. (AI052157, AI136534). J.A.T. was funded by the Hertz Graduate Fellowship. Additional funding was provided by in part by the Amy Davis Foundation, the Moore Family Foundation, and the Heidi Horner Foundation to W.A.R.

## Author contributions

J.A.T., R.M.T., W.A.R., K.L.C., and R.P.T. contributed to project conception. J.A.T., M.A.F., M.D., M.Ma., E.K., T.-S.C., A.D., and R.P.T. performed experiments and collected data. R.V.G., K.L.C., and W.A.R. consented patients. M.Ma., R.V.G., K.L.C., and W.A.R. collected, cataloged, and prepared human samples for experiments. J.A.T., M.A.F., A.D., K.L.C., J.K., and R.P.T. contributed to data analysis. J.A.T., R.M.T., M.A.F., M.Mc., R.V.G., M.Mc., A.D., W.A.R., K.L.C., J.K., and R.P.T. contributed to data interpretation. J.A.T. was the primary author. J.A.T., R.M.T., M.A.F., M.D., M.Ma., R.V.G., E.K., T.-S.C., M.Mc., A.D., W.A.R., RP, K.L.C., J.K., and R.P.T. provided revision to the scientific content of the manuscript and/or stylistic/grammatical revisions. R.M.T., A.D., W.A.R., K.L.C., M.Mc. provided funding. R.M.T., A.D., W.A.R., K.L.C., M.Mc., R.P.T., T.-S.C provided access to crucial research components. R.M.T. is the principal investigator.

## Competing interests

The authors of this paper have no competing interests to disclose.
