## [Peer Review File · Nature Communications]

Lysophosphatidic acid modulates CD8 T cell immunosurveillance and metabolism to impair anti-tumor immunityREVIEWER COMMENTS

Reviewer #1 (Remarks to the Author): expertise in immunogenomics and T cell signalling

It was previously reported that LPAR5 can act as an inhibitory receptor on T cells. In this manuscript, Turner et al. described the novel roles of LPAR signaling in CD8⁺ T cell metabolic fitness and exhaustion in antitumor immunity. Moreover, higher levels of 16:0 LPA are associated with unresponsiveness to immunotherapy in melanoma. Altogether, the authors concluded that LPAR signaling is a promising target in T cell-directed immunotherapy. The manuscript can be strengthened by addressing the following points.

- 1) Fig 1F, it is not clear which immunotherapy the patients received (anti-PD-1, anti-PD-L1, anti-CTLA4, or the combination?). The sample size here is rather small. Did the responders and non-responders receive the same treatment? And whether different treatments influence the levels of LPA.
- 2) Fig 2H, the evidence of Lpar5-mediated T cell exhaustion is weak. More exhaustion markers and T cell cytokine production should be compared between the tumor-associated WT and Lpar5^{-/-} T cells.
- 3) Fig 4 shows a dynamic response of T cells to LPA treatment. ROS can promote T cell activation in the early stage but damage T cell functions in the longer term. What will be the outcome if the LPA treatment is elongated? Is Lpar5 the main receptor of LPA in this context?
- 4) Fig 5I, there are several BODIPY dyes. Which one was used here should be mentioned. Fig 5k, 10 μ M of etomoxir was used in the Seahorse experiments to evaluate FAO. It was previously reported that the specificity of etomoxir in T cells is compromised at doses above 5 μ M and induces oxidative stress (Sci Rep 8, 6289 (2018)). Therefore, one should be careful in optimizing etomoxir concentration. Alternatively, CPT1a KO experiments could be performed to avoid an off-target effect of etomoxir.
- 5) Fig 6, a Lpar5^{-/-} OT-1 + LPA group should be added to the comparisons to assess whether LPA mainly signals through Lpar5 to modulate metabolic activities of T cells (or whether Lpar5 has an additional function in addition to its role as a receptor of LPA?)

Reviewer #2 (Remarks to the Author): expertise in CD8 T cell metabolism

In Lysophosphatidic acid modulates CD8 T cell immunosurveillance, metabolism, and anti-tumor immunity, by Turner et al, the authors explore the role of LPA and LPAR signaling on T cell function. The authors correlate LPA receptor expression with increased T cell exhaustion in human cancer and show its potential use as a biomarkers to predict response to cancer immunotherapy. The authors use a mouse model of melanoma metastasis to explore Lpar5 signaling in contributing to T cell exhaustion and analyze Lpar5 KO T cell killing in vivo, before using in vitro models to explore metabolic changes with LPA cultured T cells and the metabolic pathways induced by LPA signaling, and how that changes oxidative and glycolytic activity of T cells in vitro. These are potentially interesting findings, but as the data stands, its difficult to understand the connection between the in vitro metabolism data, the mechanism of LPA on T cell function, and its significance to T cell exhaustion in cancer. The authors need to clarify their hypothesis – does LPA signaling through Lpar5 change T cell signaling, T cell metabolism, or both? And following this, does LPA signaling or LPA-induced changes in metabolism contribute to T cell exhaustion? The in vitro results (fig 4-6) are not explored in T cell exhaustion – the authors should directly test if LPA can drive exhaustion so that we may better understand exhaustion biology and how LPAR5 may be used as a therapeutic target for cancer immunotherapy. Specific points to address are as follows:

Major points

- In figure 1A, the authors used Myc as a comparison group in their analysis of survival from multiple cancer types. Increased Myc is not associated with all cancer, so presenting the data this way is not very informative. It would be more informative if the authors showed survival data broken down by tumor type, with just high vs low Enpp2 expression. It would be interesting if Enpp2 expression was more associated with survival in a specific cancer type.
- What cells are making LPA? Can the authors use data from figure 1c to show which cells have upregulated the LPA synthesis pathway?
- Please give rationale as to why melanoma is the focus of figure 1, rather than other tumor types.
- What is tumor purity (sup fig 1D)?
- The Tim3 staining is not convincing (sup fig 3G). It looks like there is no Tim3 expressed on these cells (presumably WT OT1 cells, but is not clear). If no Tim3 is expressed on T cells in this model, the authors should not use this marker in their analysis, or use a different tumor model with Tim3 expressing T cells for analysis.
- What is the rationale for systemic B16 rather than orthotopic? Why not orthotopic as figure 1 shows a phenotype on TCGA melanoma data? Systemic B16 is more often used to model metastasis, therefore, the authors should also include orthotopic B16 modeling by injecting OT1 T cells 5-7 days after intradermal B16 implantation, and then phenotyping T cells 7-10 days later. Tumor burden may not change in the orthotopic modeling, as this is a highly aggressive tumor model, but T cell exhaustion and functionality can still be different even if no measurable change in tumor size is apparent. The author should also perform ex vivo SIINFEKL restim to quantify cytokines, as this will be more conclusive to understanding the functionality/exhaustion state of the OT1 T cells.
- Figure 3 is difficult to interpret, as full body knockout mice are used. We know from figure 1D that B cells and macrophages highly express Lpar5, so loss of Lpar5 may alter systemic immune response. The authors should remove this variable by transferring in OT1 WT or KO cells into new hosts and infect, to test T cell intrinsic killing capacity.
- The authors should make it more clear (by discussing and citing previous work in the introduction, and in the results section where relevant), specifically how Lpar5 may be functioning and how it may be impacting metabolism. The first paragraph of the introduction gives a list of enzymes Lpar5 can signal with, but its not particularly clear what the hypothesis of this work is. How is LPA signaling changing T cell function? Is it purely metabolism associated, or does it change T cell receptor signaling, or both? The previous work from this laboratory show Lpar5 change TCR signaling, but its not clear if the current hypothesis is LPAR-induced changes in TCR signaling impacts metabolism, or if LPAR-induced changes in metabolism impacts TCR signaling.
- The data from fig4A looks quite variable. The authors should plot data on PCA to determine if samples cluster together.
- What timepoint is Fig4B? Is there a difference in enrichment if different timepoints are analyzed?
- What specific BODIPY dye is used in fig 5I? There are many types of BODIPY staining specific for different lipids.
- The authors should follow up on the phenotype observed in figure 6. Does increased maximal respiration mean there are more mitochondria in these cells? Can stain with mitotracker to determine.
- In figure 6, authors are missing a group - should perform LPA treatment on Lpar5 KO cells to assess metabolism changes by LPA directly through the receptor.
- There is a disconnect from phenotype explored in figure 2 with the remaining in vitro data. How does treatment with LPA for 4hr relate to T cell exhaustion in tumors, which takes many days to occur? To connect the data more logically, the authors should add an additional figure to the manuscript where they treat cells for multiple days in vitro with LPA (either T cells cultured with LPA alone to drive exhaustion, or if LPA is insufficient by itself to drive exhaustion, then adding LPA in combination with an in vitro exhaustion assay) to see if treatment with LPA increases TCR signaling, ROS signaling, and cause T cells to become exhausted phenotypically and functionally more quickly than cells not treated with LPA. Lpar5 KO T cells cultured with LPA in an in vitro exhaustion assay to show this mitigates the negative effect of culturing T cells with LPA would also be a good control. Authors could also use Lpar5 KO T cells activated in vitro and transferred into mice with already established tumors, with or without checkpoint blockade, to determine if Lpar5 KO could be a therapeutic strategy for cellular therapies in cancer patients.

Minor points

- Can the authors comment/speculate on the role or importance of LPAR5 in other immune cells in the tumor (as shown in Fig 1C, D)?
- In some figures, text is too small to read (eg figure 4)

Reviewer #3 (Remarks to the Author): expertise in T cell mouse models

The manuscript by Turner and colleagues explores the relationship between LPAR5 signaling and T cell exhaustion. The authors demonstrated a correlation between LPA and exhausted CD8 T cells in cancer patients and in the response to immunotherapy. In vivo tumor models using LPAR5 KO mice show enhanced ability to kill tumor cells in vivo. Finally, changes in the CD8 T cell metabolism in response to LPA was shown.

Overall, the data show a correlation of LPA and an exhausted state but the data are over interpreted and over generalized at times for a largely ex vivo study that fails to validate the key findings in vivo. The experiments only coarsely evaluate exhaustion and fail to validate the metabolic findings using the in vivo models. In summary this study was largely correlative and had many more weaknesses than strengths in its current form.

Major concerns:

No data is provided in the figure legends about sample replicate numbers and how many times experiments were performed. Many of the figures (Fig 3, Fig 4I, Fig 5, Fig 6) have no mention of biological replicates or reproducibility of the data across experiments.

Authors do not integrate the findings of O'Connor et al Scientific Reports 2018 that report the detailed effects of varying concentrations of Etoximor on CD8 T cell oxidative metabolism and that at high concentrations ROS are induced.

In Figure 3, the authors appear to show how the LPAR5 signaling effects the killing ability of CD8 T cells using Lpar5^{-/-} mice. The difference that they see between the wild type and Lpar5^{-/-} mice might be caused by the effect of other immune cells that also express LPAR5 receptor.

The authors strongly claim that LPA signaling causes exhausted T cell formation by triggering metabolic changes. In Figures 5 and 6, they show that LPA signaling changes the functional metabolism and specifically that Lpar5^{-/-} mice have higher metabolic properties than WT. However, they do not provide any direct evidence that this metabolic state is linked to exhaustion or if metabolic dysfunction exists in vivo. In Fig 2 only no direct measure of exhaustion was performed beyond Lag3 and PD-1 expression.

Minor concerns:

In Figure 1D the authors should clarify and quantitative the expression of LPAR5 in distinct immune cell types to provide more robust data that the expression is enriched in CD8 T cells.

Figure 4 is low resolution and difficult to interpret panels A and B. In particular there seems to be a discrepancy between L- and D-glutamine/glutamate across the figure and text.

RESPONSE TO REVIEWERS' COMMENTS

We thank the Reviewers for their comments and helping us improve the quality of our manuscript. We believe their comments and our subsequent additional experiments and manuscript edits further the investigation of how LPA modulates CD8 T cell phenotype and function. Ultimately, we believe the Reviewers' comments and our resulting revised manuscript has not only improved our message but also strengthened our conclusions.

We have addressed each Reviewer concern point-by-point below and incorporated changes to the manuscript. Please note that references in this letter correspond to the clean version of the manuscript. We have also included a marked version of the manuscript which we submitted under optional supplemental material.

We are thankful for the opportunity to address the Reviewers' concerns. We have incorporated the recommended changes by performing the following experiments.

- 1) We performed a chronic stimulation *in vitro* assay with LPA on CD8 T cells to assess how LPA directly affects a T cell exhaustion phenotype
- 2) We designed an *in vivo* killing assay with an adoptive transfer to better assess how *Lpar5*^{-/-} on CD8 T affects antigen-specific killing
- 3) We included additional markers of T cell exhaustion for our tumor studies and further assessed CD8 T cell cytokine production *in vivo*.

In addition to these key experiments, we have taken efforts to address the Reviewers' other concerns and comments which are also incorporated throughout the manuscript.

We thank the Reviewers for these comments and feedback. We believe these changes have significantly improved the quality and impact of our manuscript.

Reviewer #1 (Remarks to the Author): expertise in immunogenomics and T cell signalling

It was previously reported that LPAR5 can act as an inhibitory receptor on T cells. In this manuscript, Turner et al. described the novel roles of LPAR signaling in CD8+ T cell metabolic fitness and exhaustion in antitumor immunity. Moreover, higher levels of 16:0 LPA are associated with unresponsiveness to immunotherapy in melanoma. Altogether, the authors concluded that LPAR signaling is a promising target in T cell-directed immunotherapy.

The manuscript can be strengthened by addressing the following points.

1) Fig 1F, it is not clear which immunotherapy the patients received (anti-PD-1, anti-PD-L1, anti-CTLA4, or the combination?). The sample size here is rather small. Did the responders and non-responders receive the same treatment? And whether different treatments influence the levels of LPA.

We agree with Reviewer #1 that this is important information that was missing, and we have updated our manuscript to include this information as Supplemental Table 2 and with patient clinical characteristics which includes the type of immunotherapies.

Each of the patients received either anti-CTLA-4, anti-PD-1, or combination anti-CTLA-4/anti-PD-1 therapy. Notably, all three responders were treated with single agent nivolumab. Interestingly, we also followed-up on the single outlier point in the non-responder group. The clinical data from patient (#9) revealed that she initially failed a combination therapy of ipilimumab/nivolumab but is now currently disease free on bispecific anti-PD-1/ICOS antibody.

We also recognize that our cohort of patients is quite small to make any definitive conclusions and thus comment on this in our discussion on page 24, paragraph 2: “We also find that elevated levels of systemically circulating LPA predict responses to immunotherapy (Figure 1F). While the cohort of patients examined is small, we did observe that plasma LPA levels predicted response to single agent nivolumab (Supplemental Table 2). There was one other patient in the cohort with low levels of LPA who did not respond to combination ipilimumab/nivolumab (Patient #9). However, after following-up on the clinical data, we found that this patient was later treated with a bispecific anti-PD-1/ICOS antibody and is currently disease free. These findings warrant further investigation into how LPAR5 functions as an immune checkpoint and may affect responses to immune checkpoint blockade with anti-CTLA4, anti-PD-1, and combination anti-CTLA4/anti-PD1 therapy.”

2) Fig 2H, the evidence of *Lpar5*-mediated T cell exhaustion is weak. More exhaustion markers and T cell cytokine production should be compared between the tumor-associated WT and *Lpar5*^{-/-} T cells.

We agree with this comment by Reviewer #1 and would like to thank the Reviewer for pointing out this weakness. As recommended, we now include additional exhaustion markers in our *in vivo* tumor model which are incorporated into a new Figure 3. We assessed expression of Tox and Lag3 on transferred CD45.1⁺ OT-I and *Lpar5*^{-/-} OT-I CD8 T cells which revealed that *Lpar5*^{-/-} OT-I CD8 T cells expressed reduced surface levels of Lag3 and Tox as compared to wild type OT-I CD8 T cells.

Further, we assessed cytokine production of the transferred CD45.1⁺ OT-I and *Lpar5*^{-/-} OT-I CD8 T cells and while the data were not statistically significant the findings trended as expected and showed that transferred CD45.1⁺ *Lpar5*^{-/-} OT-I CD8 T cells had increased production of IFN γ and TNF α . We also observed that transferred CD45.1⁺ *Lpar5*^{-/-} OT-I CD8 T cells were also trending to be more cytotoxic as measured by dual positivity for both IFN γ and CD107a. Since this data was supportive, but not definitive, we included this finding in a new Supplemental Figure 6.

We updated the manuscript text to discuss these data on page 11, paragraph 2 to say, “Using our systemic *in vivo* tumor model, we investigated additional markers of CD8 T cell exhaustion and these analyses showed that transferred CD45.1⁺ *Lpar5*^{-/-} OT-I CD8 T cells isolated from tumors in the lungs expressed reduced amounts of Lag3 and Tox as compared to wildtype transferred CD45.1⁺ OT-I CD8 T cells (Figure 3K-P).

Exhausted CD8 T cells exhibit impaired cytokine production²⁹⁻³¹ so, we also measured interferon γ (IFN γ) and tumor necrosis factor α (TNF α) production using our *in vivo* tumor model and observed that there were modest, albeit non-significant increases in dual IFN γ and TNF α production by transferred CD45.1⁺ *Lpar5*^{-/-} OT-I CD8 T cells as compared to wildtype CD45.1⁺ OT-I CD8 T cells (Supplemental Figure 6). In addition, assessing CD8 T cell cytotoxicity and function as measured by IFN γ ⁺ and surfaced CD107a⁺, we observed a supportive but non-significant trend that *Lpar5*^{-/-} OT-I CD8 T cells display increased cytotoxicity as compared to wildtype OT-I CD8 T cells. Altogether, these data provide strong evidence that *Lpar5* signaling on CD8 T cells reprograms phenotypes and increases expression of exhaustion markers both *in vitro* and *in vivo*.”

3) Fig 4 shows a dynamic response of T cells to LPA treatment. ROS can promote T cell activation in the early stage but damage T cell functions in the longer term. What will be the outcome if the LPA treatment is elongated? Is *Lpar5* the main receptor of LPA in this context?

This is a very interesting point that Reviewer #1 raises. We found previous studies showing that mitochondrial stress and ROS drive exhaustion in Tim3⁺ PD1⁺ CD8 T cells (PMID: 33398183). Thus, we thought to assess how LPA treatment may affect PD-1 and Tim3 expression *in vitro*. We performed an *in vitro* assay with chronic stimulation of wild type OT-I and *Lpar5*^{-/-} OT-I effector CD8 T cells with either LPA alone or anti-CD3+LPA and subsequently measured Tim3 and PD1 expression. We observed that there is a difference in percent of Tim3⁺ PD1⁺ CD8 T cells between wildtype OT-I or *Lpar5*^{-/-} OT-I groups. Almost the entire CD8 T cell population were positive for Tim3 and PD1 when stimulated with anti-CD3+LPA, however, in this treatment condition we observed that *Lpar5* expression modulated geometric mean fluorescence intensity of Tim3 and PD1. We updated the manuscript to include these data in Figure 3 and Supplemental Figure 4.

We report the results of these findings in the results section on page 8, paragraph 2 to say, “Since we observed decreased Tim3 expression on PD1⁺ *Lpar5*^{-/-} OT-I CD8 T cells isolated from melanoma tumors compared to wildtype OT-I CD8 T cells (Figure 2H), we sought to further investigate how LPA signaling might modulate exhausted and dysfunctional phenotypes. To accomplish this, we treated OT-I effector CD8 T cells with LPA in the presence or absence of chronic TCR stimulation *in vitro* (Figure 3A). Of note, longer-term *in vitro* cultures necessitate the use of (fetal bovine) serum which contains low levels of LPA²⁸ that likely signal via *Lpar5* throughout this culture period; nevertheless, we supplemented LPA to our cultures to ensure sustained LPA exposure for this prolonged *in vitro* assay. We also performed a chronic anti-CD3 stimulation without additional LPA supplementation, however these results were almost identical to our anti-CD3+LPA condition (data not shown). Both OT-I and *Lpar5*^{-/-} OT-I CD8 T cell cultures treated with anti-CD3+LPA resulted in virtually all CD8 T cells to dually express PD1 and Tim3 (Figure 3B-D, Supplemental Figure 4) although the level of these inhibitory receptors were reduced on *Lpar5*^{-/-} OT-I CD8 T cells (Figure 3E,H, Supplemental Figure 4). Interestingly, LPA supplementation alone in cultures of *Lpar5*^{-/-} OT-I CD8 T cells also resulted in a significantly decreased percent of PD1⁺ Tim3⁺ compared to wildtype OT-I cells (Figure 3B-D, Supplemental Figure 4). In line with our previous findings, we observed that *Lpar5*^{-/-} OT-I effector CD8 T cells that were chronically stimulated expressed less PD1 and Tim3 (Figure 3E-J). Given the robust differences we observed *in vivo*, we chose to further investigate how LPA and *Lpar5* signaling modulates exhaustion using *Lpar5* knockout mice and *in vivo* models.”

We elaborate the significance of these findings in the discussion section on page 23 paragraph 1 to say, “Considering the metabolic data in aggregate, there are strong implications for LPA modulating effector CD8 T cell production of ROS and subsequent cytotoxic function. Interestingly, previous findings show that mitochondrial stress and ROS drive exhaustion and dysfunction in Tim3⁺ PD1⁺ CD8 T cells¹¹. Considered together with our *in vitro* and *in vivo* findings assessing Tim3 and PD1 expression on wildtype OT-I and *Lpar5*^{-/-} OT-I CD8 T cells, we speculate that *Lpar5*-deficient CD8 T cells may have less ROS and mitochondrial stress which could explain our observation of decreased Tim3 and PD1 expression on *Lpar5*-deficient CD8 T cells. However, future studies are required to understand how LPAR5 and ATX signaling on CD8 T cells may affect ROS accumulation in the long-term.”

4) Fig 5l, there are several BODIPY dyes. Which one was used here should be mentioned. Fig 5k, 10 μ M of etomoxir was used in the Seahorse experiments to evaluate FAO. It was previously reported that the specificity of etomoxir in T cells is compromised at doses above 5 μ M and induces oxidative stress (Sci Rep 8, 6289 (2018)). Therefore, one should be careful in optimizing etomoxir concentration. Alternatively, CPT1a KO experiments could be performed to avoid an off-target effect of etomoxir.

As recommended, we performed a titration of etomoxir so that we may be certain that this effect was not a result of off-target side effects of etomoxir. We have updated the manuscript at Figure 6 panel K and Supplemental Figure 9.

We also updated manuscript to include what specific type of BODIPY dye was used (493/503 4,4-Difluoro-1,3,5,7,8-Pentamethyl-4-Bora-3a,4a-Diaza-s-Indacene from ThermoFisher, catalogue number: D3922) in Supplemental Table 4.

5) Fig 6, a *Lpar5*^{-/-} OT-1 + LPA group should be added to the comparisons to assess whether LPA mainly signals through *Lpar5* to modulate metabolic activities of T cells (or whether *Lpar5* has an additional function in addition to its role as a receptor of LPA?)

As recommended, we have updated Figure 7 (panels D-G) to include *Lpar5*^{-/-} OT-1 + LPA as a group to the comparisons. We believe adding this comparison group strengthens our conclusions that *Lpar5* is indeed regulating maximal respiration and proton leak but does not regulate basal respiration.

Furthermore, this comparison group also bolsters the rationale that basal metabolism is regulated by one of the other LPA receptors expressed on CD8 T cells (specifically *Lpar2* and *Lpar6*).

We wish to thank the Reviewer for this suggestion as we believe this additional comparison group has improved the quality of our data and justification for our conclusions.

Reviewer #2 (Remarks to the Author): expertise in CD8 T cell metabolism

In Lysophosphatidic acid modulates CD8 T cell immunosurveillance, metabolism, and anti-tumor immunity, by Turner et al, the authors explore the role of LPA and LPAR signaling on T cell function. The authors correlate LPA receptor expression with increased T cell exhaustion in human cancer and show its potential use as a biomarkers to predict response to cancer immunotherapy. The authors use a mouse model of melanoma metastasis to explore Lpar5 signaling in contributing to T cell exhaustion and analyze Lpar5 KO T cell killing *in vivo*, before using *in vitro* models to explore metabolic changes with LPA cultured T cells and the metabolic pathways induced by LPA signaling, and how that changes oxidative and glycolytic activity of T cells *in vitro*. These are potentially interesting findings, but as the data stands, its difficult to understand the connection between the *in vitro* metabolism data, the mechanism of LPA on T cell function, and its significance to T cell exhaustion in cancer.

The authors need to clarify their hypothesis – does LPA signaling through Lpar5 change T cell signaling, T cell metabolism, or both? And following this, does LPA signaling or LPA-induced changes in metabolism contribute to T cell exhaustion? The *in vitro* results (fig 4-6) are not explored in T cell exhaustion – the authors should directly test if LPA can drive exhaustion so that we may better understand exhaustion biology and how LPAR5 may be used as a therapeutic target for cancer immunotherapy. Specific points to address are as follows:

We thank the reviewer for highlighting this potentially confusing issue and apologize for not being clearer in our original manuscript. We have taken significant effort to address these points as described below.

We have updated our introduction on page 3, paragraph 3, to state: “Since metabolic dysfunction in CD8 T cells, impaired antigen-specific killing, and poor responses to immunotherapy are characteristics of CD8 T cells exhaustion, we hypothesized and tested whether LPA and Lpar5 signaling modulates CD8 T cell metabolic fitness and exhaustion phenotypes.”

To address this, we have generated a new figure (Figure 3) to examine how LPA and Lpar5 signaling modulate exhaustion phenotypes. We have now elongated LPA treatment on wildtype OT-I and *Lpar5*^{-/-} OT-I CD8 T cells *in vitro* to better understand how LPA and Lpar5 signaling affect exhaustion directly. We also included more markers of exhaustion in our *in vivo* tumor model and assessed cytokine production.

We expanded our discussion on the potential for future therapeutics targeting Lpar5 which could result in improved CD8 T cell metabolism during anti-tumor immune responses. Our laboratory is currently exploring these potential therapeutics (including small molecule inhibitors and monoclonal antibodies against Lpar5) in collaboration with investigators at the University of Tennessee and Massachusetts Institute of Technology, respectively.

We discuss the need for future studies to better examine the role of proton leak in CD8 T cells. A recent study published by Nature in 2022 (PMID: 35614225) challenges our fundamental and dogmatic understanding of proton leak. Altogether, proton leak is incompletely understood and that there is a need to define the exact mechanism of proton leak, *especially* in CD8 T cells. We have updated our manuscript to emphasize this point and highlight the cutting-edge insight that our paper adds to the existing body of knowledge around proton leak and oxidative damage in CD8 T cells. Specifically, we discuss on page 23, paragraph 2 that “In sum, our data shows Lpar5 signaling fluxes ROS, results in lipid peroxidation, and increases proton leak. While metabolic state and CD8 T cell exhaustion are highly associated, it is unclear whether the metabolic state induced by LPA alone can drive CD8 T cell exhaustion. It could be possible that LPA drives generalized oxidative damage and targeting metabolism could be a future avenue for investigation. Notably, our current understanding of the mechanisms regulating lipotoxicity, oxidative stress, and proton leak are actively being investigated and evolving⁴⁶⁻⁴⁹. A recent publication has challenged our fundamental and dogmatic understanding of proton uncoupling⁴⁶. Importantly, metabolic pathways that result in proton uncoupling, leak, and oxidative damage have been reported to be a key fate-determining mechanism in T cells^{11,13,50,51}. Thus, our data contribute to an emerging and important field in immunometabolism which highlight the need for future studies to elucidate the specific mechanisms of how oxidative damage and proton leak are regulated in CD8 T cells.”

Major points

- In figure 1A, the authors used Myc as a comparison group in their analysis of survival from multiple cancer types. Increased Myc is not associated with all cancer, so presenting the data this way is not very informative. It would be more informative if the authors showed survival data broken down by tumor type, with just high vs low Enpp2 expression. It would be interesting if Enpp2 expression was more associated with survival in a specific cancer type.

We believe that this is an important point that Reviewer #2 raised. We have stratified our data by cancer type to better understand this effect and updated the manuscript to include this information in Supplemental Figure 1.

We have several cancer types stratified and the overlap of cancer types harboring an *ENPP2* or *MYC* amplifications was sporadic and there were only a few cancer types in which we could investigate with robust numbers. However, tumor types with insufficient numbers were grouped into an 'All Other Cancers' subset for additional analysis.

Interestingly, with the data that we have now stratified, we found that patients with all other cancers and genitourinary cancers exhibit worse progression survival with *ENPP2* amplification, while those with endometrial carcinoma and serous ovarian cancer exhibit better survival.

- What cells are making LPA? Can the authors use data from figure 1c to show which cells have upregulated the LPA synthesis pathway?

We thank Reviewer #2 for commenting on this. We have added to Supplemental Figure 1 to address this point. Since we know that extracellular LPA synthesis depends on *ENPP2* expression, we plotted *ENPP2* expression and updated Supplemental Figure 1 with these data. We have updated the text of the manuscript to reference this at Page 5, paragraph 2. We further note in the introduction on page 3, paragraph 1 that, as a bioactive signaling molecule, LPA is generated extracellularly through the enzymatic action of the *ENPP2* gene product, autotaxin. Thus, cells expressing *ENPP2* are expected to be actively generating extracellular LPA.

- Please give rationale as to why melanoma is the focus of figure 1, rather than other tumor types.

We apologize that we did not clarify this point and have updated the manuscript to better explain our rationale for focusing on melanoma, which had been used previously by our group and remains a focus for our attention.

Specifically, on page 5, paragraph 2 we say, "Previously, we have shown that LPAR5-deficient CD8 T cells are better able to kill melanoma tumor cells *in vitro* and control local tumor growth after implantation and compared to wildtype CD8 T cells^{8,16}. The first report of ATX generating LPA was first identified in melanoma and our laboratory has established interest in examining the role of ATX in melanoma²⁴, as such, we specifically focused on this cancer type for our study."

Importantly, this is not to say that ATX does not play key roles in other tumor types and our findings very well could potentially be applied to other cancers. In the discussion on page 25, paragraph 2 we explain the LPA has been reported to play key roles in the development and progression of other cancers including prostate, breast, ovarian, etc. We expanded on this point to say, "Future studies should investigate how LPA, ATX, and LPAR5 signaling modules CD8 T cell phenotypes in these cancer types."

- What is tumor purity (sup fig 1D)?

We apologize for our incomplete explanation of tumor purity. Sequencing analyses from TIMER2.0 assesses tumor purity to inform on density of tumor cells versus other cells in the microenvironment. In Supplemental Figure 1D we observed an inverse relationship with *Lpar5* expression and tumor purity which suggest that

LPAR5 is not expressed on tumor cells. Specifically, samples that have pure tumor show decreased LPAR5 since there are fewer immune cells within bulk tumor sample and as the purity of tumor cells decrease and immune cell infiltrate increases, the expression of LPAR5 also increases. These data serve to support the other findings from Figure 1 which show that LPAR5 is expressed on lymphocytes including CD8 T cells.

- The Tim3 staining is not convincing (sup fig 3G). It looks like there is no Tim3 expressed on these cells (presumably WT OT1 cells, but is not clear). If no Tim3 is expressed on T cells in this model, the authors should not use this marker in their analysis, or use a different tumor model with Tim3 expressing T cells for analysis.

We apologize for this poor representation of this flow cytometric data. We have updated relevant figures with a better representative image and clarified which sample the plot was taken from and included this in the figure legend.

- What is the rationale for systemic B16 rather than orthotopic? Why not orthotopic as figure 1 shows a phenotype on TCGA melanoma data? Systemic B16 is more often used to model metastasis, therefore, the authors should also include orthotopic B16 modeling by injecting OT1 T cells 5-7 days after intradermal B16 implantation, and then phenotyping T cells 7-10 days later. Tumor burden may not change in the orthotopic modeling, as this is a highly aggressive tumor model, but T cell exhaustion and functionality can still be different even if no measurable change in tumor size is apparent. The author should also perform *ex vivo* SIINFEKL restim to quantify cytokines, as this will be more conclusive to understanding the functionality/exhaustion state of the OT1 T cells.

This is a very interesting point that Reviewer #2 raised. We agree with the Reviewer and we performed an orthotopic B16 *in vivo* experiment. As suggested, we performed intradermal injection to implant B16.cOVA orthotopically, adoptively transferred OT-I or *Lpar5*^{-/-} OT-I CD8 T cells on day 10 and further allowed tumor growth for an additional 7 days (17 days after implantation).

Since we did not observe any significant differences in tumor volume or expression of PD1, Lag3, Tox, Tim3, or TCF1 on CD8 T cells in either wildtype OT-I or *Lpar5*^{-/-} OT-I recipient mice, we did not move forward to assess cytokines using this model but included these data as Supplemental Figure 5.

Importantly, we agree with Reviewer #2 that cytokine production from *ex vivo* SIINFEKL restimulation is an important assessment of CD8 T cell function/exhaustion. As such, we performed this *ex vivo* SIINFEKL stimulation experiment in our systemic tumor model and included as Supplemental Figure 6. We discuss this data in our results section on page 11, paragraph 2 as: "Exhausted CD8 T cells exhibit impaired cytokine production²⁹⁻³¹ so, we also measured interferon γ (IFN γ) and tumor necrosis factor α (TNF α) production using our *in vivo* tumor model and observed that there were modest, albeit non-significant increases in dual IFN γ and TNF α production by transferred CD45.1⁺ *Lpar5*^{-/-} OT-I CD8 T cells as compared to wildtype CD45.1⁺ OT-I CD8 T cells (Supplemental Figure 6). In addition, assessing CD8 T cell cytotoxicity and function as measured by IFN γ ⁺ and surfaced CD107a⁺, we observed a supportive but non-significant trend that *Lpar5*^{-/-} OT-I CD8 T cells display increased cytotoxicity as compared to wildtype OT-I CD8 T cells."

- Figure 3 is difficult to interpret, as full body knockout mice are used. We know from figure 1D that B cells and macrophages highly express *Lpar5*, so loss of *Lpar5* may alter systemic immune response. The authors should remove this variable by transferring in OT1 WT or KO cells into new hosts and infect, to test T cell intrinsic killing capacity.

We agree with Reviewer #2 that other cell types could be contributing to killing effect we observe in Figure 4. As recommended, we performed an *in vivo* killing assay with an adoptive transfer of wildtype OT-I or *Lpar5*-deficient OT-I CD8 T cells. We chose to adoptively transfer 10,000 cells to allow us to clearly assess the

response from the transferred cells with effect observed from the endogenous cells responding (PMID: 17555991).

We adoptively transferred *Lpar5*^{-/-} OT-I CD8 T cells or wildtype OT-I CD8 T cells and then immunized the mice with OVA peptide. Similar to our previous experiment, we transferred pulsed and unpulsed target cells on Day 5. To best visualize the difference in antigen-specific killing, we harvested the mice 2 hours after transferring the target cells and observed a significant difference in antigen-specific killing *in vivo*. We updated the manuscript with these findings in Figure 4 and discussed it in the results on page 12, paragraph 2: “Since B cells and macrophages also express *LPAR5* (Figure 1D), we sought to assess the CD8 T cell-specific contribution to antigen-specific killing *in vivo* and performed this experiment with an adoptive transfer of wildtype OT-I or *Lpar5*^{-/-} OT-I CD8 T cells (Figure 4F). Using this adoptive transfer model, and consistent with findings in Figure 4E, we observed that mice transferred with *Lpar5*^{-/-} OT-I CD8 T cells exhibit improved antigen-specific killing *in vivo* as compared to mice transferred with wildtype OT-I CD8 T cells (Figure 4G-H).”

- The authors should make it more clear (by discussing and citing previous work in the introduction, and in the results section where relevant), specifically how *Lpar5* may be functioning and how it may be impacting metabolism. The first paragraph of the introduction gives a list of enzymes *Lpar5* can signal with, but its not particularly clear what the hypothesis of this work is. How is LPA signaling changing T cell function? Is it purely metabolism associated, or does it change T cell receptor signaling, or both? The previous work from this laboratory show *Lpar5* change TCR signaling, but its not clear if the current hypothesis is LPAR-induced changes in TCR signaling impacts metabolism, or if LPAR-induced changes in metabolism impacts TCR signaling.

We apologize for the confusing hypothesis and as recommended, we have amended our introduction on page 3, paragraph 3 to say, “Since metabolic dysfunction in CD8 T cells, impaired antigen-specific killing, and poor responses to immunotherapy are characteristics of CD8 T cells exhaustion, we hypothesized and tested whether LPA and *Lpar5* signaling modulates CD8 T cell metabolic fitness and exhaustion phenotypes.”

We discuss the specific idea that oxidative damage (possibly induced by LPA) could determine cell fate in the discussion on page 23, paragraph 2: “In sum, our data shows *Lpar5* signaling fluxes ROS, results in lipid peroxidation, and increases proton leak. While metabolic state and CD8 T cell exhaustion are highly associated, it is unclear whether the metabolic state induced by LPA alone can drive CD8 T cell exhaustion. It could be possible that LPA drives generalized oxidative damage and targeting metabolism could be a future avenue for investigation. Notably, our current understanding of the mechanisms regulating lipotoxicity, oxidative stress, and proton leak are actively being investigated and evolving⁴⁶⁻⁴⁹. A recent publication has challenged our fundamental and dogmatic understanding of proton uncoupling⁴⁶. Importantly, metabolic pathways that result in proton uncoupling, leak, and oxidative damage have been reported to be a key fate-determining mechanism in T cells^{11,13,50,51}. Thus, our data contribute to an emerging and important field in immunometabolism which highlight the need for future studies to elucidate the specific mechanisms of how oxidative damage and proton leak are regulated in CD8 T cells.”

- The data from fig4A looks quite variable. The authors should plot data on PCA to determine if samples cluster together.

As recommended, we have performed PCA analyses to show that the samples do indeed cluster together. We have included this piece of data as Supplemental Figure 8E-F.

- What timepoint is Fig4B? Is there a difference in enrichment if different timepoints are analyzed?

This enrichment analysis isn't based on a single timepoint but was rather generated using *metaboAnalyst* software which integrates each group and compares enrichment differences across all groups. We have

clarified this analysis in the text in the methods section on page 27, paragraph 1 to say, "Data analysis was performed using MetaboAnalyst and the enrichment analysis was performed to integrate comparisons across all groups."

- What specific BODIPY dye is used in fig 5I? There are many types of BODIPY staining specific for different lipids.

We thank Reviewer #2 for this commenting on this. We used BODIPY 493/503 (4,4-Difluoro-1,3,5,7,8-Pentamethyl-4-Bora-3a,4a-Diaza-s-Indacene) from ThermoFisher (catalogue number: D3922). We have updated our Supplemental Table 4 to include this information.

- The authors should follow up on the phenotype observed in figure 6. Does increased maximal respiration mean there are more mitochondria in these cells? Can stain with mitotracker to determine.

As recommended, we followed up on these findings and better characterized the mitochondria in these cells. We performed mitotracker staining and included this data as Supplemental Figure 10.

We updated the results section of manuscript to comment on these findings on page 20, paragraph 1: "Since total amount of mitochondria could affect maximal respiratory capacity, we performed MitoTracker staining as a semi-quantitative measure of mitochondrial mass. We observed a subtle and transient decrease in MitoTracker in both the wildtype OT-I and *Lpar5*^{-/-} OT-I effector CD8 T cells treated with 1 μ M LPA (Supplemental Figure 10A-D). However, we did not observe a significant difference in total mitochondrial mass between wildtype OT-I and *Lpar5*^{-/-} OT-I effector CD8 T cells in the absence of LPA (Supplemental Figure 10E-F)."

Based the temporal nature of these findings, we included a comment in the discussion on page 22, paragraph 3 to say, "Our data revealed that LPA induced a rapid depletion of neutral lipids in effector CD8 T cells. Triglycerides from neutral lipids are catabolized to free fatty acids which are shuttled into the mitochondria for oxidative consumption⁴⁰. However, if triglycerides are broken down from lipid droplets and not used for energetic consumption, then these free fatty acids can become lipotoxic in the cytosol. Previous groups have reported that lipid droplets may serve a protective role by buffering cellular amounts of toxic lipids that cause oxidative stress and lipotoxicity^{41,42}. Our results show LPA modulates oxidative stress in CD8 T cells via a metabolic mechanism. Specifically, we observed LPA signaling results in a transient flux of H₂O₂ while at the same time a corresponding increase in lipid peroxidation (Figure 5H,J). We did not observe an increase in lipid peroxidation in *Lpar5*^{-/-} OT-I effector CD8 T cells (Figure 5K). Considered together, our data shows evidence that oxidative damage and lipotoxicity is a consequence of *Lpar5* signaling. Yet, we also observed that both wildtype OT-I and *Lpar5*^{-/-} OT-I effector CD8 T cells treated with LPA exhibit a transient decrease in mitochondrial mass (Supplemental Figure 10 A-D). This transient decrease is subtle, and the exact biological significance of this modulation remains unclear."

- In figure 6, authors are missing a group - should perform LPA treatment on *Lpar5* KO cells to assess metabolism changes by LPA directly through the receptor.

We thank the Reviewer for pointing this out and we have updated our groups in the figure, now Figure 7, to include LPA treatment on *Lpar5*^{-/-} cells (panels D-G).

As referenced above in a point #5 from Reviewer #1, we are very grateful for this comment as we believe including this additional comparison strengthens our conclusions that *Lpar5* regulates maximal respiratory capacity and proton leak but not basal respiration.

• There is a disconnect from phenotype explored in figure 2 with the remaining *in vitro* data. How does treatment with LPA for 4hr relate to T cell exhaustion in tumors, which takes many days to occur? To connect the data more logically, the authors should add an additional figure to the manuscript where they treat cells for multiple days *in vitro* with LPA (either T cells cultured with LPA alone to drive exhaustion, or if LPA is insufficient by itself to drive exhaustion, then adding LPA in combination with an *in vitro* exhaustion assay) to see if treatment with LPA increases TCR signaling, ROS signaling, and cause T cells to become exhausted phenotypically and functionally more quickly than cells not treated with LPA. *Lpar5* KO T cells cultured with LPA in an *in vitro* exhaustion assay to show this mitigates the negative effect of culturing T cells with LPA would also be a good control. Authors could also use *Lpar5* KO T cells activated *in vitro* and transferred into mice with already established tumors, with or without checkpoint blockade, to determine if *Lpar5* KO could be a therapeutic strategy for cellular therapies in cancer patients.

We thank the Reviewer for bringing this important point to our attention. As recommended, we included a new figure (Figure 3) in the main manuscript to further investigate the relationship between LPA, *Lpar5*, and exhaustion. We first measured markers of exhaustion in wildtype OT-I or *Lpar5*^{-/-} OT-I CD8 T cells cultured *in vitro* with LPA or anti-CD3+LPA to better assess how LPA may directly affect expression of exhaustion markers. We further investigated how *Lpar5* modulates exhausted phenotypes using our *in vivo* tumor model to assess more markers of exhaustion (specifically Lag3 and Tox in Figure 3) and cytokine production (Supplemental Figure 6). These new results reveal that *Lpar5* signaling promotes exhaustive differentiation and dysfunction.

We updated the body of the text to include a new section on page 8, paragraph 2: “Since we observed decreased Tim3 expression on PD1⁺ *Lpar5*^{-/-} OT-I CD8 T cells isolated from melanoma tumors compared to wildtype OT-I CD8 T cells (Figure 2H), we sought to further investigate how LPA signaling might modulate exhausted and dysfunctional phenotypes. To accomplish this, we treated OT-I effector CD8 T cells with LPA in the presence or absence of chronic TCR stimulation *in vitro* (Figure 3A). Of note, longer-term *in vitro* cultures necessitate the use of (fetal bovine) serum which contains low levels of LPA²⁸ that likely signal via *Lpar5* throughout this culture period; nevertheless, we supplemented LPA to our cultures to ensure sustained LPA exposure for this prolonged *in vitro* assay. We also performed a chronic anti-CD3 stimulation without additional LPA supplementation, however these results were almost identical to our anti-CD3+LPA condition (data not shown). Both OT-I and *Lpar5*^{-/-} OT-I CD8 T cell cultures treated with anti-CD3+LPA resulted in virtually all CD8 T cells to dually express PD1 and Tim3 (Figure 3B-D, Supplemental Figure 4) although the level of these inhibitory receptors were reduced on *Lpar5*^{-/-} OT-I CD8 T cells (Figure 3E,H, Supplemental Figure 4). Interestingly, LPA supplementation alone in cultures of *Lpar5*^{-/-} OT-I CD8 T cells also resulted in a significantly decreased percent of PD1⁺ Tim3⁺ compared to wildtype OT-I cells (Figure 3B-D, Supplemental Figure 4). In line with our previous findings, we observed that *Lpar5*^{-/-} OT-I effector CD8 T cells that were chronically stimulated expressed less PD1 and Tim3 (Figure 3E-J). Given the robust differences we observed *in vivo*, we chose to further investigate how LPA and *Lpar5* signaling modulates exhaustion using *Lpar5* knockout mice and *in vivo* models.

Previously, our laboratory has shown *Lpar5*^{-/-} CD8 T cells impede local tumor growth better than wildtype CD8 T cells^{8,16}. Thus, we first used an orthotopic tumor model to investigate how *Lpar5* modulates CD8 T cell exhaustion, however, we did not observe any significant difference in CD8 T cell exhaustion markers in this model (Supplemental Figure 5). Using our systemic *in vivo* tumor model, we investigated additional markers of CD8 T cell exhaustion and these analyses showed that transferred CD45.1⁺ *Lpar5*^{-/-} OT-I CD8 T cells isolated from tumors in the lungs expressed reduced amounts of Lag3 and Tox as compared to wildtype transferred CD45.1⁺ OT-I CD8 T cells (Figure 3K-P). Exhausted CD8 T cells exhibit impaired cytokine production²⁹⁻³¹ so, we also measured interferon γ (IFN γ) and tumor necrosis factor α (TNF α) production using our *in vivo* tumor model and observed that there were modest, albeit non-significant increases in dual IFN γ and TNF α production by transferred CD45.1⁺ *Lpar5*^{-/-} OT-I CD8 T cells as compared to wildtype CD45.1⁺ OT-I CD8 T cells (Supplemental Figure 6). In addition, assessing CD8 T cell cytotoxicity and function as measured by IFN γ ⁺ and surfaced CD107a⁺, we observed a supportive but non-significant trend that *Lpar5*^{-/-} OT-I CD8 T cells display increased cytotoxicity as compared to wildtype OT-I CD8 T cells. Altogether, these data provide strong

evidence that Lpar5 signaling on CD8 T cells reprograms phenotypes and increases expression of exhaustion markers both *in vitro* and *in vivo*.”

Minor points

- Can the authors comment/speculate on the role or importance of LPAR5 in other immune cells in the tumor (as shown in Fig 1C, D)?

As recommended, we included more discussion on LPAR5 in other immune cells. We discussed some of the previously published data from the Torres Laboratory that examined LPA signaling on B cells and human T cells in addition to other future project directions including the role of LPAR5 on NK and macrophages. Specifically, we updated the manuscript on page 24, paragraph 2: “Previously, our laboratory has reported that LPA signaling through Lpar5 on B cells and T cells impairs intracellular calcium signaling downstream of the T cell and B cell receptors^{16,53}. Interestingly, macrophages and NK cells also express LPAR5 (Figure 1C-D), yet the exact role and function of LPAR5 on myeloid and other lymphoid lineage cell types remains poorly defined.”

- In some figures, text is too small to read (eg figure 4)

We apologize for this small text and have amended our figure so that it can be more easily read.

Reviewer #3 (Remarks to the Author): expertise in T cell mouse models

The manuscript by Turner and colleagues explores the relationship between LPAR5 signaling and T cell exhaustion. The authors demonstrated a correlation between LPA and exhausted CD8 T cells in cancer patients and in the response to immunotherapy. *In vivo* tumor models using LPAR5 KO mice show enhanced ability to kill tumor cells *in vivo*. Finally, changes in the CD8 T cell metabolism in response to LPA was shown. Overall, the data show a correlation of LPA and an exhausted state but the data are over interpreted and over generalized at times for a largely *ex vivo* study that fails to validate the key findings *in vivo*. The experiments only coarsely evaluate exhaustion and fail to validate the metabolic findings using the *in vivo* models. In summary this study was largely correlative and had many more weaknesses than strengths in its current form.

Major concerns:

No data is provided in the figure legends about sample replicate numbers and how many times experiments were performed. Many of the figures (Fig 3, Fig 4I, Fig 5, Fig 6) have no mention of biological replicates or reproducibility of the data across experiments.

We apologize for this. We have updated our figure legends and we also verified each of these points on the Nature Portfolio reporting summary for rigor and reproducibility which has been sent to the Editorial team at Nature Communications.

Authors do not integrate the findings of O'Connor et al Scientific Reports 2018 that report the detailed effects of varying concentrations of Etoximir on CD8 T cell oxidative metabolism and that at high concentrations ROS are induced.

We thank Reviewer #3 for their comment on this and Reviewer #1 also had a similar concern. To address the reviewer concerns we performed a titration curve of etomoxir and updated Figure 6 and Supplemental Figure 9 with these data.

In Figure 3, the authors appear to show how the LPAR5 signaling effects the killing ability of CD8 T cells using *Lpar5*^{-/-} mice. The difference that they see between the wild type and *Lpar5*^{-/-} mice might be caused by the effect of other immune cells that also express LPAR5 receptor.

We agree with Reviewer #3. A similar comment was shared with Reviewer #2. To address these Reviewer concerns, we performed this *in vivo* killing assay with an adoptive transfer to better study the effect *Lpar5*^{-/-} on the CD8 T cells. As also discussed above in point #7 from Reviewer #2, we observed a significant difference in antigen-specific killing *in vivo* using an adoptive transfer of either wildtype OT-I or *Lpar5*^{-/-} OT-I CD8 T cells *in vivo* antigen-specific killing assay. We updated the manuscript with these findings in Figure 4 and discussed it in the results on page 12, paragraph 2: "Since B cells and macrophages also express LPAR5 (Figure 1D), we sought to assess the CD8 T cell-specific contribution to antigen-specific killing *in vivo* and performed this experiment with an adoptive transfer of wildtype OT-I or *Lpar5*^{-/-} OT-I CD8 T cells (Figure 4F). Using this adoptive transfer model, and consistent with findings in Figure 4E, we observed that mice transferred with *Lpar5*^{-/-} OT-I CD8 T cells exhibit improved antigen-specific killing *in vivo* as compared to mice transferred with wildtype OT-I CD8 T cells (Figure 4G-H)."

The authors strongly claim that LPA signaling causes exhausted T cell formation by triggering metabolic changes. In Figures 5 and 6, they show that LPA signaling changes the functional metabolism and specifically that *Lpar5*^{-/-} mice have higher metabolic properties than WT. However, they do not provide any direct

evidence that this metabolic state is linked to exhaustion or if metabolic dysfunction exists *in vivo*. In Fig 2 only no direct measure of exhaustion was performed beyond Lag3 and PD-1 expression.

We apologize for the confusion around our hypothesis and experimental design. We aimed to test the hypothesis that LPA regulates antigen-specific killing and metabolic fitness. We further examined how Lpar5 signaling drives exhaustion phenotypes. We clarified our introduction to better state what we are specifically testing on page 3, paragraph 3: “Since metabolic dysfunction in CD8 T cells, impaired antigen-specific killing, and poor responses to immunotherapy are characteristics of CD8 T cells exhaustion, we hypothesized and tested whether LPA and Lpar5 signaling modulates CD8 T cell metabolic fitness and exhaustion phenotypes.”

We further agree with Reviewer #3 that the connection between exhaustion and Lpar5 signaling can be strengthened. As recommended, we have included more in-depth *in vitro* and *in vivo* experiments are incorporated in the manuscript as Figure 3 and Supplemental Figures 4 and 6. Initially, we measured Tim3 and PD1 expression using our *in vivo* tumor model described in Figure 2. We have now provided more direct evidence that Lpar5 signaling modulates exhaustion phenotypes by:

- 1) Performing an *in vitro* assay to culture wildtype OT-I and *Lpar5*^{-/-} OT-I CD8 T cells with LPA in the presence and absence of TCR stimulation. We observed that fewer PD1⁺ Tim3⁺ when cultured with LPA alone in *Lpar5*-deficient T cells and further, chronic stimulation resulted in decreased expression of PD1 and Tim3 in *Lpar5*-deficient T cells (Figure 3A-J and Supplemental Figure 4).
- 2) Assessing more markers of exhaustion on transferred CD45.1⁺ OT-I and *Lpar5*^{-/-} OT-I CD8 T cells using our *in vivo* tumor model (Figure 3J-O).
- 3) Measuring cytokine production in transferred CD45.1⁺ OT-I and *Lpar5*^{-/-} OT-I CD8 T cells from our *in vivo* tumor model (Supplemental Figure 6).

Altogether, we believe that these provide convincing evidence that Lpar5 regulates exhaustion phenotypes.

We have updated our manuscript text to describes these data on page 8, paragraph 2: “Since we observed decreased Tim3 expression on PD1⁺ *Lpar5*^{-/-} OT-I CD8 T cells isolated from melanoma tumors compared to wildtype OT-I CD8 T cells (Figure 2H), we sought to further investigate how LPA signaling might modulate exhausted and dysfunctional phenotypes. To accomplish this, we treated OT-I effector CD8 T cells with LPA in the presence or absence of chronic TCR stimulation *in vitro* (Figure 3A). Of note, longer-term *in vitro* cultures necessitate the use of (fetal bovine) serum which contains low levels of LPA²⁸ that likely signal via Lpar5 throughout this culture period; nevertheless, we supplemented LPA to our cultures to ensure sustained LPA exposure for this prolonged *in vitro* assay. We also performed a chronic anti-CD3 stimulation without additional LPA supplementation, however these results were almost identical to our anti-CD3+LPA condition (data not shown). Both OT-I and *Lpar5*^{-/-} OT-I CD8 T cell cultures treated with anti-CD3+LPA resulted in virtually all CD8 T cells to dually express PD1 and Tim3 (Figure 3B-D, Supplemental Figure 4) although the level of these inhibitory receptors were reduced on *Lpar5*^{-/-} OT-I CD8 T cells (Figure 3E,H, Supplemental Figure 4). Interestingly, LPA supplementation alone in cultures of *Lpar5*^{-/-} OT-I CD8 T cells also resulted in a significantly decreased percent of PD1⁺ Tim3⁺ compared to wildtype OT-I cells (Figure 3B-D, Supplemental Figure 4). In line with our previous findings, we observed that *Lpar5*^{-/-} OT-I effector CD8 T cells that were chronically stimulated expressed less PD1 and Tim3 (Figure 3E-J). Given the robust differences we observed *in vivo*, we chose to further investigate how LPA and Lpar5 signaling modulates exhaustion using Lpar5 knockout mice and *in vivo* models.

Previously, our laboratory has shown *Lpar5*^{-/-} CD8 T cells impede local tumor growth better than wildtype CD8 T cells^{8,16}. Thus, we first used an orthotopic tumor model to investigate how Lpar5 modulates CD8 T cell exhaustion, however, we did not observe any significant difference in CD8 T cell exhaustion markers in this model (Supplemental Figure 5). Using our systemic *in vivo* tumor model, we investigated additional markers of CD8 T cell exhaustion and these analyses showed that transferred CD45.1⁺ *Lpar5*^{-/-} OT-I CD8 T cells isolated from tumors in the lungs expressed reduced amounts of Lag3 and Tox as compared to wildtype transferred CD45.1⁺ OT-I CD8 T cells (Figure 3K-P). Exhausted CD8 T cells exhibit impaired cytokine production²⁹⁻³¹ so, we also measured interferon γ (IFN γ) and tumor necrosis factor α (TNF α) production using our *in vivo* tumor

model and observed that there were modest, albeit non-significant increases in dual IFN γ and TNF α production by transferred CD45.1⁺ *Lpar5*^{-/-} OT-I CD8 T cells as compared to wildtype CD45.1⁺ OT-I CD8 T cells (Supplemental Figure 6). In addition, assessing CD8 T cell cytotoxicity and function as measured by IFN γ ⁺ and surfaced CD107a⁺, we observed a supportive but non-significant trend that *Lpar5*^{-/-} OT-I CD8 T cells display increased cytotoxicity as compared to wildtype OT-I CD8 T cells. Altogether, these data provide strong evidence that Lpar5 signaling on CD8 T cells reprograms phenotypes and increases expression of exhaustion markers both *in vitro* and *in vivo*.”

Importantly, we also discuss the leading edge of the field where future hypotheses can be formed and studies conducted. Specifically, on page 23, paragraph 2 we update our discussion to say, “In sum, our data shows Lpar5 signaling fluxes ROS, results in lipid peroxidation, and increases proton leak. While metabolic state and CD8 T cell exhaustion are highly associated, it is unclear whether the metabolic state induced by LPA alone can drive CD8 T cell exhaustion. It could be possible that LPA drives generalized oxidative damage and targeting metabolism could be a future avenue for investigation. Notably, our current understanding of the mechanisms regulating lipotoxicity, oxidative stress, and proton leak are actively being investigated and evolving⁴⁶⁻⁴⁹. A recent publication has challenged our fundamental and dogmatic understanding of proton uncoupling⁴⁶. Importantly, metabolic pathways that result in proton uncoupling, leak, and oxidative damage have been reported to be a key fate-determining mechanism in T cells^{11,13,50,51}. Thus, our data contribute to an emerging and important field in immunometabolism which highlight the need for future studies to elucidate the specific mechanisms of how oxidative damage and proton leak are regulated in CD8 T cells.”

Lastly, we want to make a point to thank Reviewer #3 for this feedback and sincerely believe the manuscript modifications we have made to address this point have clarified our hypotheses, strengthened the justification for our conclusions, and increased the impact of our manuscript. In our first response to Reviewer #2, we discuss the important need to define the exact mechanism of proton leak, *specifically* in CD8 T cells. We have updated our manuscript to highlight this and emphasize that our work adds to the leading edge of the immunometabolism field and inform future directions to better explore oxidative damage as a fate-directing phenomenon.

Minor concerns:

In Figure 1D the authors should clarify and quantitative the expression of LPAR5 in distinct immune cell types to provide more robust data that the expression is enriched in CD8 T cells.

To address this point, we included a more in-depth analysis of LPAR5 expression across the distinct immune cell types in Supplemental Figure 1.

Figure 4 is low resolution and difficult to interpret panels A and B. In particular there seems to be a discrepancy between L- and D-glutamine/glutamate across the figure and text.

We apologize for the low resolution, and we have updated the figure and we reviewed the text for discrepancies between L- and D-glutamine/glutamate. Of note, there are instances where we measured D-glutamine/glutamate and other times where we supplemented the media with L-glutamine. As such, we carefully reviewed the text to be sure that we are accurately describing what we measured and supplemented with.

REVIEWERS' COMMENTS

Reviewer #1 (Remarks to the Author):

The authors have addressed all of my concerns and suggestions in a satisfactory manner. The revised manuscript now presents a more robust and convincing study.

Reviewer #2 (Remarks to the Author):

Lysophosphatidic acid modulates CD8 T cell immunosurveillance, metabolism, and anti-tumor immunity by Turner et al., the authors made a commendable effort to address the concerns raised in review. Unfortunately, the new data do not show the link between T cell exhaustion, a phenomenon that takes many days to occur, and the mechanism explored in 4hr timecourses in the 2nd half of the paper. With 3 new and different experiments showing no difference in T cell cytokine functionality (sup fig5, Fig3A-J, and sup fig 6), it's difficult to conclude that LPAR5 signaling has a significant impact on T cell exhaustion. Co-inhibitory marker differences and Tox differences could simply be showing us the KO T cells are less differentiated, not that they are less exhausted. Without this essential piece of cell functionality data, I do not recommend this manuscript for publication in its current form.

Reviewer #3 (Remarks to the Author):

The authors have adequately responded to all of my comments. The results are a much improved manuscript and interpretation of the data.